# Loss of ALK4 promotes cancer progression through regulating TGF-β receptor N-glycosylation

Manqi Zhang[1,7], Jian Chen[1,7], Mary C. Leupold [1], Jennifer Guo[1], Xiangfeng Shen[2], Carl A. Shirley[3], Christen A. Khella [4,5], Rushabh N. Mehta[4,5], Edward T. O'Brien[1], Michael L. Gatza [4,5] & Gerard C. Blobe [1,6] ✉

The transforming growth factor-β (TGF-β) pathway typically inhibits tumorigenesis but can promote metastasis during cancer progression. Activin receptor-like kinase 4 (ALK4), a type I TGF-β family receptor, is frequently downregulated or mutated in cancers, and reduced ALK4 expression correlates with poorer outcomes. However, its role and mechanism of action in cancer progression remains unclear. We demonstrate that ALK4 loss enhances anchorage-independent growth, migration, invasion, and epithelial-mesenchymal transition in vitro, as well as cancer progression in breast and pancreatic cancer models in vivo. Importantly, ALK4 loss promotes canonical TGF-β signaling by increasing TGF-β receptor N-linked glycosylation and stabilizing these receptors at the cell surface. Mechanistically, ALK4 loss upregulates β1,6 N-acetylglucosaminyltransferase V (MGAT5) and galectin-3, which binds MGAT5-modified glycoproteins to stabilize surface receptors. Consistent with prior observations that galectin-3 preferentially binds to MGAT5-modified glycoproteins to stabilize cell surface receptors like TGF-β receptors, we demonstrate that ALK4 loss enhances MGAT5-mediated glycosylation of TGF-β receptors, promoting their stabilization and signal transduction. Depleting MGAT5 or inhibiting N-glycosylation effectively suppresses ALK4-loss-induced TGF-β signaling and cancer progression.

The transforming growth factor-β (TGF-β) family, which includes TGF-βs, activins, growth and differentiation factors (GDFs), bone morphogenetic proteins, and inhibins, plays essential roles in embryonic development and regulates key cellular processes such as proliferation, differentiation, apoptosis, and migration. These proteins signal by binding and activating their respective cell surface receptors[1–4].

TGF-β signaling begins with TGF-β binding to the type III TGF-β receptor (TβRIII), which facilitates ligand presentation to the type II TGF-β receptor (TβRII)[5,6]. This interaction recruits the type I TGF-β receptor (TβRI) to form a receptor complex. TβRII phosphorylates and activates TβRI[7–9], which subsequently phosphorylates the transcription factors Smad2 and/or Smad3. The activated Smads complex with Smad4 translocates to the nucleus, where they regulate

[1]Department of Medicine, Division of Medical Oncology, Duke University Medical Center, Durham, NC, USA. [2]Head and Neck Surgery & Communications Sciences, Duke University, Durham, NC, USA. [3]Department of Dermatology, University of Wisconsin, Madison, WI, USA. [4]Rutgers Cancer Institute of New Jersey, New Brunswick, NJ, USA. [5]Department of Radiation Oncology, Robert Wood Johnson Medical School, Rutgers, The State University of New Jersey, New Brunswick, NJ, USA. [6]Department of Pharmacology and Cancer Biology, Duke University Medical Center, Durham, NC, USA. [7]These authors contributed equally: Manqi Zhang, Jian Chen. ✉e-mail: gerard.blobe@duke.edu

the transcription of TGF-β target genes. TGF-β signaling has a dichotomous role in cancer progression. In early stages, it suppresses tumorigenesis by inhibiting proliferation and promoting differentiation and apoptosis[8,9]. Conversely, in advanced stages, elevated TGF-β signaling drives epithelial-to-mesenchymal transition (EMT), immune evasion, chemotherapy resistance, and metastasis[10–13], making TGF-β signaling an attractive therapeutic target.

Common strategies to inhibit TGF-β signaling include blocking ligand–receptor interaction and inhibiting receptor kinase activity[14]. Although preclinical studies show potent antitumor effects of TGF-β-targeting agents[15–17], clinical outcomes have been less encouraging. To effectively suppress TGF-β signaling and prevent cancer, the mechanisms regulating this pathway must be fully understood. Another approach focuses on reducing TGF-β1 responsiveness, including promoting endocytosis to destabilize cell surface receptors, a process regulated by β1,6 N-acetylglucosaminyltransferase V (MGAT5)-glycosylation[18]. Notably, MGAT5 depletion in cancer models reduces metastasis and cell proliferation[19–21]. Furthermore, understanding factors that regulate TGF-β signaling is crucial for selecting patients who may benefit from TGF-β inhibition and for identifying biomarkers to assess treatment efficacy[6].

Activin-like kinase 4 (ALK4), encoded by *ACVR1B*, is a type I receptor that primarily mediates activin A signaling. Loss of heterozygosity and somatic biallelic inactivation of the *ACVR1B* locus have been observed in pancreatic cancer cell lines and clinical samples[22,23]. A Sleeping Beauty Transposon Screen identified *Acvr1b* as a gene whose disruption, in combination with oncogenic KRAS, promotes pancreatic adenocarcinoma in vivo[24]. Consistent with this, genetically engineered mouse models show that ALK4 deletion in combination with oncogenic KRAS drives pancreatic tumorigenesis[25,26], and CRISPR-Cas9-mediated ALK4 depletion enhances colon cancer tumorigenicity in the presence of APC and KRAS mutations[27]. Although ALK4 loss has been linked to pancreatic and colon cancer, its role and mechanism in cancer progression remain poorly understood.

Here, we investigate and identify a mechanism by which ALK4 loss enhances TGF-β signaling and promotes cancer progression in pancreatic and triple-negative breast cancers, offering new insights into potential therapeutic approaches targeting TGF-β signaling pathways in these cancer types.

## Table 1 | Mutation and copy number variation of ALK4 in human cancers

| Cancer type | Mutation frequency (COSMIC) | Copy number variation (TCGA) |
|---|---|---|
| Breast | 57/3150 (1.81%) | Loss 147/963 (15.26%) |
| Basal-like breast cancer | 3/166 (1.80%) | Loss 95/166 (57.2%) |
| Cervix | 5/335 (1.49%) | Loss 15/278 (5.40%) |
| CNS | 12/2755 (0.44%) | Loss 40/283 (14.13%) |
| Endometrium | 40/813 (4.92%) | Loss 25/549 (5.28%) |
| Colon | 110/3051 (3.61%) | Loss 23/220 (10.45%) |
| Esophagus | 41/1467 (2.79%) | Loss 60/265 (22.64%) |
| Head and neck | NA | Loss 51/496 (10.28%) |
| Kidney | 12/2472 (0.49%) | Loss 4/446 (0.90%) |
| Liver | 51/2315 (2.2%) | Loss 37/360 (10.28%) |
| Lung | 52/3121 (1.67%) | Loss 47/230 (20.43%) |
| Melanoma | NA | Loss 60/287 (20.90%) |
| Ovary | 10/1025 (0.98%) | Loss 54/311 (17.36%) |
| Pancreas | 76/2177 (3.49%) | Loss 37/109 (33.95%) |
| Prostate | 24/2049 (1.17%) | Loss 23/492 (4.67%) |
| Stomach | 41/1467 (2.79%) | Loss 47/369 (12.74%) |

## Results

### Low *ACVR1B* expression correlates with poor outcomes in breast and pancreatic cancer

We initially assessed alterations in the *ACVR1B* gene in human cancer patient cohorts. Mutation rates and loss of copy number variation for *ACVR1B* across cancer types ranged from 0.44% to 4.92% and from 0.90% to 33.95%, respectively (Table 1). Notably, both breast and pancreatic cancers exhibited high mutation and copy number loss rates for *ACVR1B*. *ACVR1B* mRNA levels were significantly reduced in breast and pancreatic tumors compared to normal tissues (Fig. 1a, b). Consistent with these data, we found that *ACVR1B* expression was significantly lower in primary pancreatic tumors relative to adjacent normal tissue (Fig. 1c).

Low *ACVR1B* expression was significantly associated with reduced recurrence-free survival in breast cancer patients (Fig. 1d) and with significantly decreased overall survival in pancreatic cancer patients (Fig. 1e, Supplementary Fig. 1a). Moreover, reduced *ACVR1B* expression correlated with poorer recurrence-free survival in breast cancer patients with high Ki67 proliferation indices (Supplementary Fig. 1b). Importantly, *ACVR1B* expression was significantly lower in basal-like breast cancer compared to Luminal A (LumA), LumB, and HER2-enriched subtypes (Fig. 1f). Furthermore, lower *ACVR1B* expression was associated with worse recurrence-free survival in basal-like breast cancer patients (Fig. 1g). Consistent with patient data, ALK4 expression at the mRNA level was more frequently decreased in basal-like breast cancer cell lines, including the MDA-MB-231 cell line, compared to luminal breast cancer cell lines (Supplementary Fig. 2a). ALK4 expression was particularly low in the MDA-MB-231 LM2 subline, a derivative selected for increased lung metastatic capability in vivo[28] (Supplementary Fig. 2b). These findings highlight the potential of *ACVR1B* as a prognostic marker for poor outcomes in both breast and pancreatic cancers. They also suggest that the loss of ALK4 expression is a common event in these cancers and may contribute to their progression.

The downregulation of *ACVR1B* in tumor samples is, at least in part, attributed to epigenetic regulation, particularly DNA methylation, a well-established mechanism for silencing gene expression in cancer. In both breast and pancreatic cancer cells, we observed a significant negative correlation between *ACVR1B* expression and *ACVR1B* methylation levels (Supplementary Fig. 2c, d), suggesting that *ACVR1B* methylation is a critical mechanism driving *ACVR1B* silencing. Notably, the MDA-MB-231 LM2 subline exhibited higher *ACVR1B* methylation levels than the parental MDA-MB-231 cells (Supplementary Fig. 2e). Treatment of LM2 cells with 5-Azacitidine (5-AZA), a demethylating agent, significantly increased *ACVR1B* expression (Supplementary Fig. 2f). Similarly, in pancreatic cancer, tumor samples exhibited significantly higher *ACVR1B* methylation levels compared to adjacent normal tissues (Supplementary Fig. 2g), and 5-AZA treatment also significantly upregulated *ACVR1B* expression in pancreatic cancer cells (Supplementary Fig. 2h).

### Loss of ALK4 promotes cancer progression

We investigated the functional role of ALK4 in cancer progression using in vivo models. Silencing ALK4 expression in MDA-MB-231 cells (Supplementary Fig. 3a) significantly increased pulmonary lesion formation following tail vein injection into immunodeficient mice (Fig. 2a). This effect was reversed by re-expressing a shRNA-resistant ALK4 construct (Supplementary Fig. 3a, Fig. 2a). Conversely, restoring ALK4 expression in the MDA-MB-231 LM2 subline (Supplementary Fig. 3b) significantly reduced pulmonary lesion formation (Fig. 2b–e) and prolonged overall survival (Fig. 2f). Similarly, restoring ALK4 expression in the 4T1 mammary carcinoma line (Supplementary Fig. 3c) did not affect primary orthotopic tumor growth (Supplementary Fig. 3d, e). However, it significantly suppressed the development

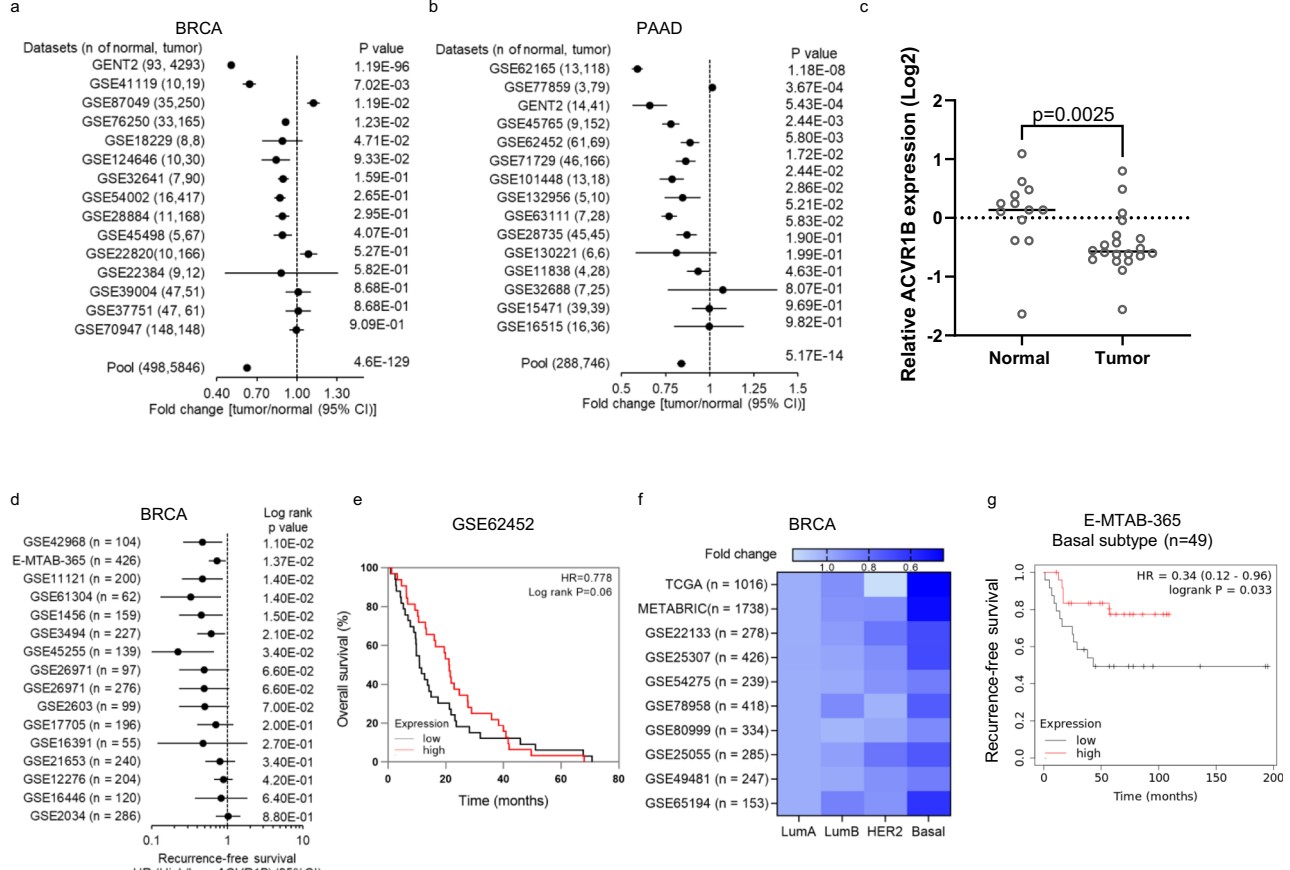

**Fig. 1 | Low *ACVR1B* expression correlates with poorer clinical outcomes in pancreatic and breast cancer patients.** Forest plots with fold changes in *ACVR1B* expression between primary breast cancers (BRCA) and normal breast tissue (**a**) and primary pancreatic cancer (PAAD) and normal pancreatic tissue (**b**) for the indicated datasets. Dots represent fold changes, and horizontal lines indicate 95% confidence intervals (CI). **c** Relative *ACVR1B* mRNA expression (Log2) in normal adjacent pancreatic (*n* = 13) and primary pancreatic cancer tissues (*n* = 20), quantitated by qRT-PCR. *ACVR1B* expression was normalized to *GAPDH* and compared to the mean of normal adjacent pancreas samples. **d** Forest plot of hazard ratios (HRs)

for recurrence-free survival in BRCA patients with high (above the median) versus low (below the median) *ACVR1B* expression in the indicated cohorts. Dots represent HRs, and horizontal lines indicate the 95% CIs. **e** Kaplan–Meier plot of overall survival for PAAD patients stratified by median *ACVR1B* expression (data from GSE62452, *n* = 68). **f** Association between *ACVR1B* expression and BRCA PAM50 subtype classification in the indicated cohorts. **g** Kaplan–Meier plot of recurrence-free survival for BRCA patients with basal-like subtype, stratified by median *ACVR1B* expression (data from E-MTAB-365, *n* = 49). Data were analyzed using the nonparametric two-tailed Mann–Whitney test.

of spontaneous pulmonary metastases (Supplementary Fig. 3f–h) and extended overall survival (Supplementary Fig. 3i). Similarly, in the HPNE orthotopic pancreatic cancer model, silencing ALK4 reduced primary tumor growth (Fig. 2g, h, Supplementary Fig. 3j) but increased the incidence of both local invasiveness (Fig. 2i) and distant metastasis (Fig. 2j–k).

To further elucidate the impact of ALK4 during cancer progression, we utilized an autochthonous mouse model. In Kras$^{LSL-G12D}$;Acvr1b$^{fl/fl}$;Pdx1-Cre (AKC) mice, ALK4 depletion in the KRAS$^{G12D}$-expressing pancreata significantly reduced overall survival compared to Kras$^{LSL-G12D}$;Pdx1-Cre (KC) controls (Supplementary Fig. 4a). Consistent with previous studies[25,29], ALK4 depletion in the KRAS$^{G12D}$-expressing pancreas significantly promoted pancreatic cyst formation (Supplementary Fig. 4b). By 8–9 months of age, the pancreatic weight of AKC mice was significantly higher than that of KC mice, with no significant differences in body weight (Supplementary Fig. 4c, d). Notably, all AKC mice at 8–9 months of age developed pancreatic ductal adenocarcinoma (PDAC) with variable degrees of differentiation, whereas most KC mice exhibited low- or high-grade pancreatic intraepithelial neoplasias (PanINs), of which only 3/10 progressed to PDAC (Supplementary Fig. 4b, e; Table 2). These in vivo studies demonstrated that loss of ALK4 promotes cancer progression.

## ALK4 loss promotes epithelial-to-mesenchymal transition, migration, and invasion

We next investigated how loss of ALK4 contributes to cancer progression. Silencing ALK4 in MDA-MB-231 and PANC-1 cells (Supplementary Figs. 3a and 5a–b) induced EMT (Fig. 3a, b, Supplementary Fig. 5c), a pattern also observed in MCF10A cells (Supplementary Fig. 5d–f). Consistent with EMT induction, ALK4 loss significantly enhanced anchorage-independent growth (Fig. 3c), cell migration (Fig. 3d–f), invasion (Fig. 3g, h), and the formation of invasive structures in 3D collagen assays (Supplementary Fig. 5g). Restoring ALK4 expression in MDA-MB-231 LM2 cells reversed these changes, reducing mesenchymal marker expression (Supplementary Fig. 5h), cell migration (Fig. 3i), invasion (Fig. 3j), and invasive structure formation (Supplementary Fig. 5i, j). These effects were independent of cell proliferation, as ALK4 loss showed neutral to suppressive effects on proliferation (Supplementary Fig. 6). Mechanistically, increased directional persistence partially accounted for the enhanced migration following ALK4 loss (Fig. 3k, l).

The in vitro findings were corroborated by in silico data. Using data from TCGA, CPTAC-PAAD, QCMG-PAAD, and METABRIC-BRCA, we stratified patients with pancreatic cancer (PAAD) and breast cancer (BRCA) into top and bottom quartiles of *ACVR1B* expression and then

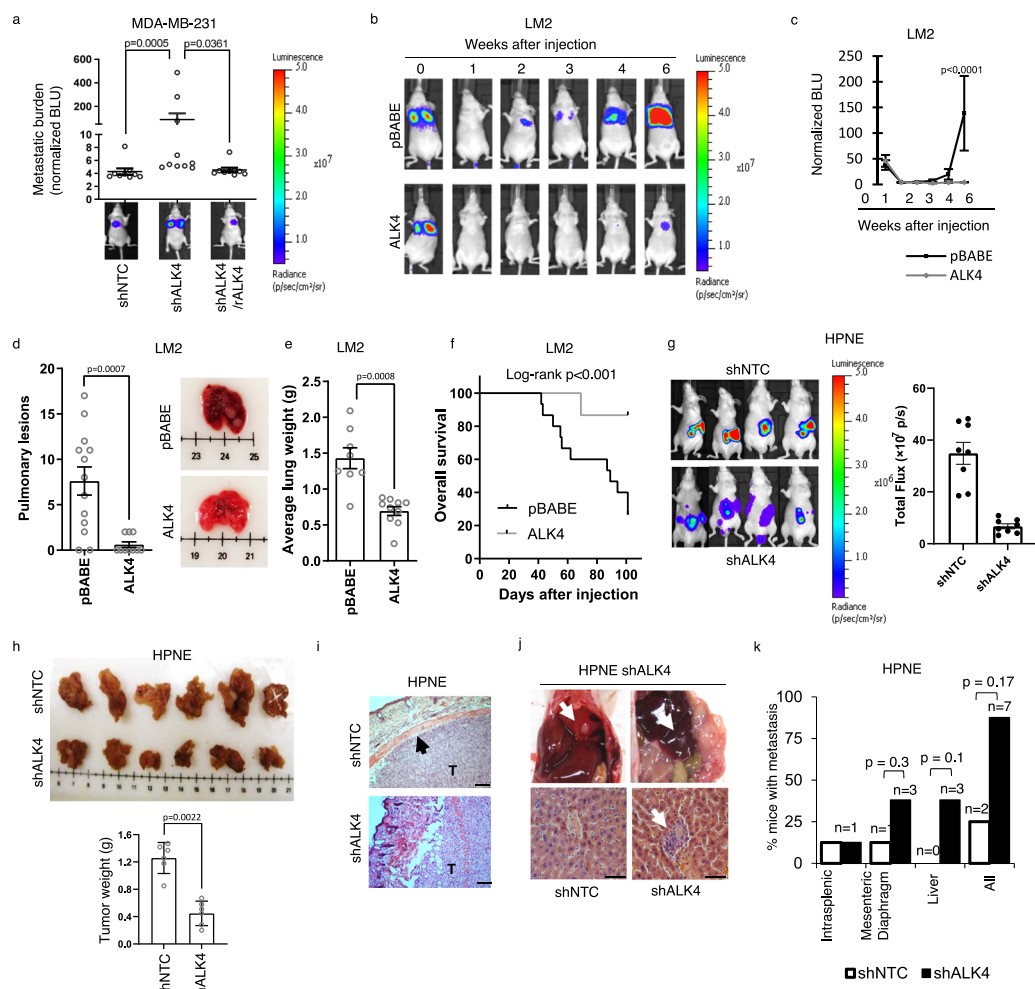

**Fig. 2 | Loss of ALK4 promotes cancer progression. a** MDA-MB-231 cells stably expressing a control shRNA (shNTC), shRNA targeting ALK4 (shALK4), or shRNA-resistant ALK4 (rALK4) were injected into the tail vein of nude mice ($n = 10$ per group). Pulmonary lesions were detected by bioluminescent imaging. Total bioluminescence at the end of the experiment (week 7) is presented with representative images. Statistical analysis was performed using nonparametric one-way ANOVA (Kruskal–Wallis test) followed by Dunn's multiple comparisons test. Data are presented as mean values ± SEM. **b–f** LM2 cells expressing pBABE or pBABE-ALK4 vectors were injected into the tail vein of nude mice ($n = 15$ per group). Pulmonary lesions were imaged by bioluminescent imaging at the indicated times. **b** Representative images of LM2 pBABE- and pBABE-ALK4-injected mice. **c** Normalized bioluminescence of pulmonary lesions at the indicated times. Data are presented as mean values ± SD. Data were analyzed by ordinary two-way ANOVA with Tukey's multiple comparisons test. **d** Average number of palpable pulmonary lesions with representative images of the lungs. Data are presented as

mean values ± SEM. **e** Average lung weight at week 7 of LM2 pBABE- and pBABE-ALK4-injected mice. Data are presented as mean values ± SEM. **f** Kaplan–Meier survival curves for LM2 pBABE- and pBABE-ALK4-injected mice. **g–k** HPNE pancreatic cells stably expressing shNTC or shALK4 were orthotopically injected into the tail of mouse pancreata ($n = 8$ per group). **g** Total bioluminescence and representative images of the mice 7 weeks after injection. **h** Primary pancreatic tumors from mice injected with HPNE cells expressing shNTC or shALK4 were dissected and weighed. **i** H&E staining of primary pancreatic tumors from mice bearing HPNE cells expressing shNTC or shALK4. Scale bar = 150 μm. **j** Representative images of spontaneous metastases observed in mice bearing orthotopic HPNE pancreatic cells expressing shALK4. Scale bar = 50 μm. **k** Quantification of spontaneous metastases in mice bearing orthotopic HPNE cells expressing shNTC or shALK4, data were analyzed by a two-sided Chi-square test. Other data comparing two groups were analyzed using a nonparametric two-tailed Mann–Whitney test.

performed gene set enrichment analysis (GSEA) using EMT and cancer-associated gene signatures. We observed significant enrichment of multiple gene signatures of EMT and invasive cancer phenotypes in the ALK4 low patient group (Fig. 3m, n), underscoring the clinical relevance of these phenotypes. Together, these results establish a role for ALK4 loss in driving EMT, migration, and invasion in both breast and pancreatic cancers.

## ALK4 loss promotes TGF-β signaling
ALK4, a type I receptor for activin A, also mediates signaling by Nodal and GDF5 ligands[9,30,31]. To explore the mechanism underlying ALK4 loss-driven cancer progression, we examined its effect on signaling pathways activated by these ligands. While reducing ALK4 expression did not consistently alter activin A- or Nodal-induced Smad2/3

phosphorylation, it significantly enhanced TGF-β1-mediated Smad2/3 phosphorylation in both breast and pancreatic cancer models (Fig. 4a–c). In addition to increasing Smad2/3 phosphorylation, ALK4 loss also promoted nuclear accumulation of phospho-Smad2 following TGF-β1 stimulation (Fig. 4d) and upregulated TGF-β target genes in PANC-1 and MDA-MB-231 cells (Supplementary Fig. 7a, b). MDA-MB-231 cells lacking ALK4 exhibited TGF-β-responsive reporter gene expression at levels comparable to those induced by exogenous TGF-β (Supplementary Fig. 7c). In vivo, orthotopic injection of HPNE cells with ALK4 knockdown into mice corroborated these findings, showing elevated phospho-Smad2 levels in the absence of ALK4 expression (Supplementary Fig. 7d).

Consistent with our in vitro findings, we observed that low ALK4 expression strongly correlated with active TGF-β signaling, whereas

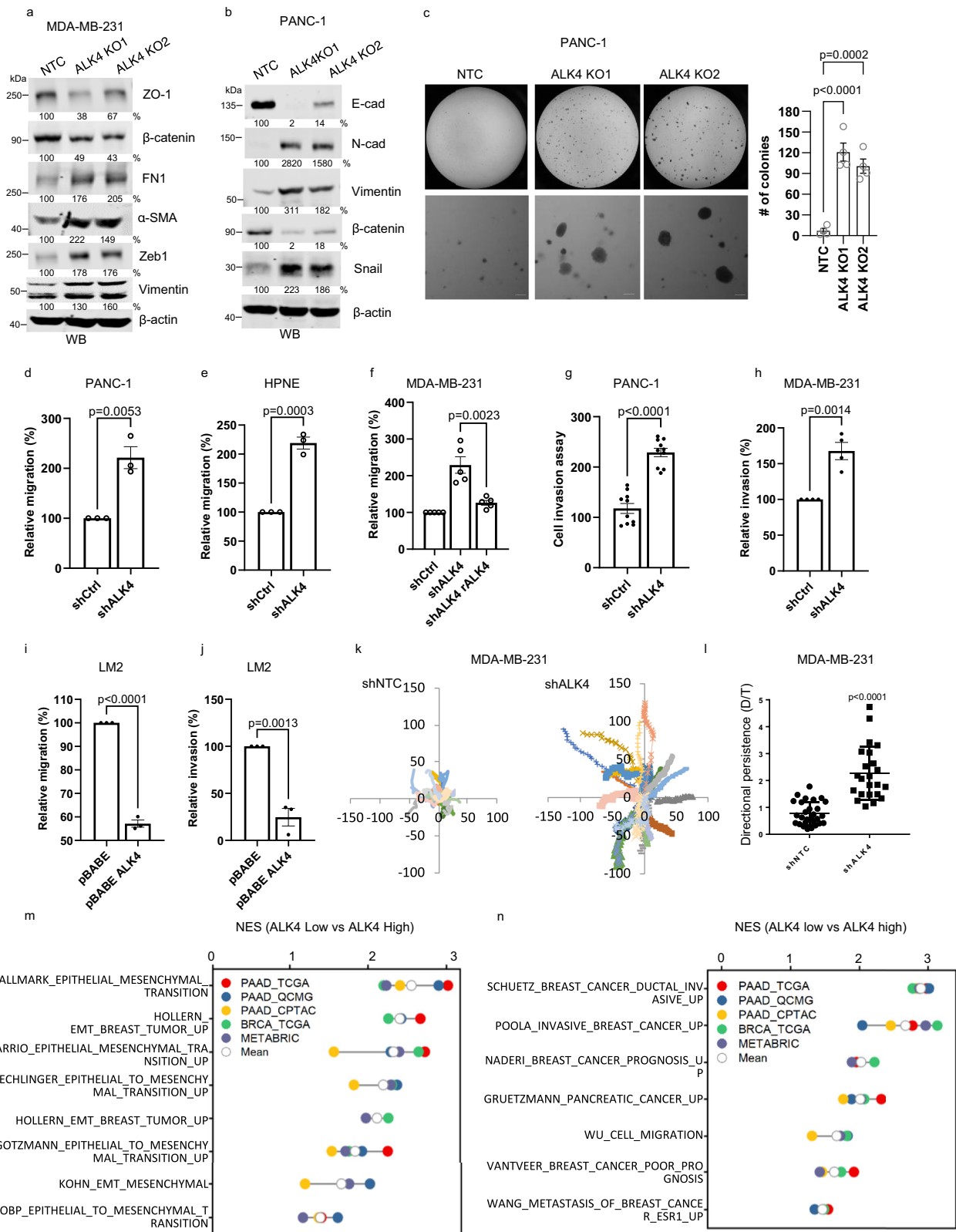

high ALK4 expression inversely correlated with TGF-β signaling in breast and pancreatic cancer specimens (Fig. 4e). Specifically, reduced ALK4 expression correlated with increased expression of TGF-β target genes, including key effectors and mediators of EMT (Fig. 4f, g, Supplementary Fig. 7e, f). Single-cell RNA sequencing (scRNA-seq) data from breast cancer[32] further revealed lower ALK4 expression in basal-like breast tumor cells, with reduced ALK4 expression negatively

correlating with TGF-β-regulated gene expression (Supplementary Fig. 7g).

To determine whether the downstream effects of ALK4 loss were driven by enhanced TGF-β signaling, we inhibited TGF-β signaling using a dominant-negative type II TGF-β receptor (DN-TβRII) (Supplementary Fig. 8a). This effectively reversed both the increase in TGF-β signaling (Fig. 4h, i) and migration induced by ALK4 loss (Fig. 4j).

**Fig. 3 | Loss of ALK4 promotes EMT, anchorage-independent growth, migration, and invasion. a** MDA-MB-231 and **b** PANC-1 CRISPR non-targeting control (NTC) cells and two CRISPR ALK4 KO clones for each cell line were analyzed for expression of EMT markers and β-actin by Western blotting. One representative out of three independent replicates is shown. **c** Soft agar colony formation assay of PANC-1 NTC cells and two ALK4 KOs clones ($n = 4$ independent experiments), scale bar = 200 nm. Data were analyzed using ordinary one-way ANOVA followed by Dunnett's multiple comparisons test. Cell migration was assessed by transwell migration assay in PANC-1 (**d**) and HPNE (**e**) cells expressing shNTC or shALK4 ($n = 3$ biological replicates). **f** MDA-MB-231 cells expressing shNTC, shALK4, or shALK4 and shRNA-resistant ALK4 ($n = 5$ biological replicates) were analyzed using transwell migration assays. Cell invasion was assessed using Matrigel transwell assays in PANC-1 ($n = 10$ biological replicates) (**g**) and MDA-MB-231($n = 4$ biological replicates) (**h**) cells expressing shNTC or shALK4. LM2 cells expressing pBABE or pBABE-ALK4 ($n = 3$ biological replicates) were analyzed for migration (**i**) and invasion (**j**). **k** MDA-MB-231 cells expressing shNTC or shALK4 were sparsely seeded and subjected to live cell imaging to track migration. Data from 23 cells for each line are shown, with directional persistence calculated in (**l**). Pancreatic (PAAD) and breast (BRCA) cancer patient data from TCGA, CPTAC-PAAD, QCMG-PAAD, and METABRIC-BRCA were stratified based on ALK4 expression. Patients in the bottom quartile (low ALK4) and top quartile (high ALK4) of ALK4 expression from these cohorts were selected. Gene expression data were used for gene set enrichment analysis (GSEA) with cancer-associated gene signatures related to EMT (**m**) or invasive cancer phenotypes (**n**) that were significantly enriched in the ALK4-low group. Normalized enrichment scores (NES) with $p$-values < 0.05 are indicated by the colored symbols, as defined in the plot legends. The soft agar, migration, and invasion assays were quantified in a blinded manner. Data comparing two groups were analyzed using two-tailed Student's $t$-tests. Data are presented as mean values ± SEM.

## Table 2 | Histological evaluation of pancreatic sections of GEM mice

| Animal group | Tumor classification | Details |
|---|---|---|
| AKC_1 | PDAC | Intermediate differentiation |
| AKC_2 | PDAC | Well differentiated |
| AKC_3 | PDAC | Intermediate differentiation |
| AKC_4 | PDAC | Poorly differentiated |
| AKC_5 | PDAC | Well differentiated |
| AKC_6 | PDAC | IPMN component, well differentiated |
| AKC_7 | PDAC | Well differentiated |
| AKC_8 | PDAC | Intermediate differentiation |
| AKC_9 | PDAC | Well differentiated |
| AKC_10 | PDAC | IMPN component, well differentiated |
| KC_1 | PanIn | High grade |
| KC_2 | PanIn | High grade |
| KC_3 | PanIn | High grade |
| KC_4 | PanIn | Low grade |
| KC_5 | PDAC | Intermediate differentiation |
| KC_6 | PDAC | Well differentiated |
| KC_7 | PanIn | High grade with acinar ductal metaplasia and chronic pancreatitis |
| KC_8 | PanIn | Low-grade with acinar ductal metaplasia and chronic pancreatitis |
| KC_9 | PDAC | Intermediate differentiation, see notes |
| KC_10 | PanIn | High grade with acinar ductal metaplasia and chronic pancreatitis |

Similarly, silencing the type I TGF-β receptor (TβRI/ALK5) through siRNA or expressing a dominant-negative TβRI blocked the increased migration and TGF-β signaling caused by ALK4 loss (Supplementary Fig. 8b–d).

Next, we examined whether ALK4 loss affects TGF-β signaling through suppression of activin A signaling. Using sActRIIB-Fc, a soluble activin receptor that sequesters activin A[33], we assessed whether this could replicate the effects of ALK4 loss. While sActRIIB-Fc inhibited activin A-induced Smad2 phosphorylation (Supplementary Fig. 9a), sActRIIB-Fc did not recapitulate the effects of loss of ALK4, including inducing TGF-β-induced Smad2 phosphorylation and promoting EMT (reduction of E-cadherin, induction of Slug and α-SMA) (Supplementary Fig. 9b). Instead, sActRIIB-Fc promoted an epithelial phenotype (decreased Slug and increased E-cadherin), suggesting that ALK4 loss mediates its effects on TGF-β signaling and EMT independently of activin signaling suppression.

## ALK4 loss increases TβRI and TβRII surface expression and complex formation via regulated receptor glycosylation and stability

To explore the mechanism by which ALK4 loss enhances TGF-β signaling, we examined the effect of ALK4 expression on TβRI or TβRII at the mRNA level, finding no significant changes (Supplementary Fig. 10). We then assessed whether ALK4 loss promotes TGF-β signaling by increasing the cell surface expression of TGF-β receptors. Using an $I^{125}$-TGF-β binding and crosslinking assay, we found that ALK4 loss led to increased surface expression of TβRI and TβRII in breast and pancreatic cancer cells (Fig. 5a, b, Supplementary Fig. 11a, b). This was further confirmed by cell surface biotinylation assay (Supplementary Fig. 11c, d). Since TβRI cannot bind I125-TGF-β without forming a complex with ligand-bound TβRII[34], increased levels of I125-TGF-β-bound TβRI with loss of ALK4 also support increased complex formation between TβRII and TβRI in this context. Conversely, overexpressing wild-type ALK4 or constitutively active ALK4 significantly reduced the surface expression of TβRI and TβRII, whereas kinase-inactive ALK4 had no such effect (Fig. 5c, d), suggesting that ALK4's kinase activity is critical for regulating TβRI/TβRII cell surface expression. Additionally, restoring ALK4 expression in ALK4-deficient LM2 breast cancer cells decreased both TβRI and TβRII surface expression and complex formation (Fig. 5e).

Receptor surface expression is often regulated by glycosylation in the Golgi apparatus[35,36]. Consistently, mutations in the N-glycosylation sites of TβRII significantly impaired TGF-β-induced Smad2 phosphorylation and downstream signaling[37]. Overexpressing active ALK4 reduced TβRII glycosylation and shifted most TβRII to a less- or non-glycosylated form (Fig. 5f, Supplementary Fig 11e), which was confirmed by PNGase F treatment that removes all N-linked glycosylation (Fig. 5f). In contrast, silencing ALK4 increased the levels of both glycosylated and non-glycosylated TβRII in pancreatic and breast cancer models and enhanced resistance to endoglycosidase H (EndoH) (Fig. 5g), supporting preferential accumulation of glycosylated TβRII. Supporting this, N-glycosylation inhibition with tunicamycin reduced TβRI and TβRII surface expression (Fig. 5h) and attenuated the effects of ALK4 loss on TβRII expression (Fig. 5i).

## Proteomic analysis reveals potential mechanisms by which ALK4 loss promotes TGF-β signaling and cancer progression

To assess how ALK4 loss enhances TGF-β signaling, we conducted proteomic analyses of whole-cell lysates from Panc1 control (NTC) and two independent Panc1 ALK4 CRISPR knockout (KO) lines. Both ALK4 KO lines exhibited similar patterns of differentially expressed proteins (DEPs) (Fig. 6a, Supplementary Fig. 12a). Functional annotation and pathway enrichment analyses using Database for Annotation, Visualization, and Integrated Discovery (DAVID) of the most upregulated DEPs revealed a strong association between ALK4 loss and N-linked

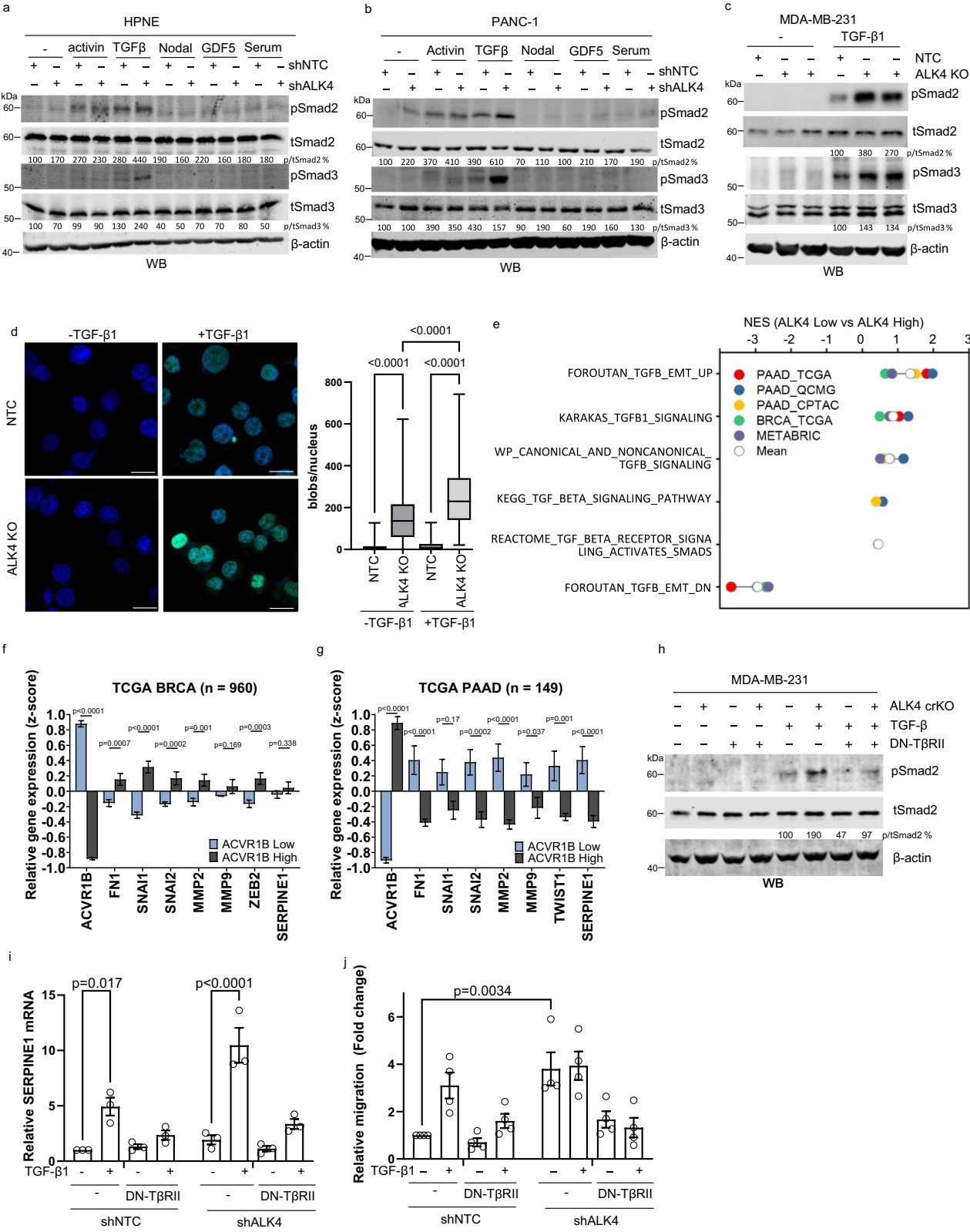

glycosylation, with many of these proteins localized to the plasma membrane and extracellular region (Fig. 6b, Supplementary Fig. 12b). These proteins were enriched in pathways related to TGF-β signaling, integrin-mediated signaling, and extracellular matrix organization (Fig. 6b).

Consistent with our previously described findings linking ALK4 loss to enhanced EMT, GSEA identified positive enrichment of EMT and

cell migration pathways in ALK4 KO cells (Fig. 6c, Supplementary Fig. 12c). Furthermore, genes commonly upregulated in pancreatic cancer were significantly enriched in ALK4 KO cells (Supplementary Fig. 12d). Multiple TGF-β signaling signatures correlated positively with ALK4 KO cells (Fig. 6d, e, Supplementary Fig. 12e), whereas retained ALK4 expression was associated with the downregulation of a TGF-β EMT signature (Fig. 6f).

**Fig. 4 | Loss of ALK4 promotes TGFβ signaling.** HPNE (**a**) and PANC-1 (**b**) cells expressing shNTC or shALK4 were serum-starved for 3 h and then treated with vehicle, 100 pM activin A, 100 pM TGF-β1, 100 ng/ml Nodal, 1 nM GDF5, or 5% serum for 30 min. Protein lysates were analyzed by Western blotting. One representative out of three independent replicates is shown. **c** MDA-MB-231 NTC cells and two isogenic ALK4 KO lines were serum-starved for 3 h and then treated with vehicle or 100 pM TGF-β1 for 30 min. Protein lysates were analyzed by Western blotting. One representative out of three independent replicates is shown. **d** Nuclear localization of pSmad2 in NTC-TGF-β1 ($n = 165$ cells), NTC + TGF-β1 ($n = 324$ cells), ALK4 KO-TGF-β1 ($n = 93$ cells), and ALK4 KO + TGF-β1 ($n = 194$ cells) groups was assessed by immunofluorescent microscopy. Scale bar = 25 μm. Nuclear blob counts were quantified using BlobFinder v3.2. The central line marks the median, the box extends from the 25th to the 75th percentiles. The whiskers go from min to max. **e** Pancreatic (PAAD) and breast (BRCA) cancer patients from the TCGA, CPTAC-PAAD, QCMG-PAAD, and METABRIC-BRCA cohorts were stratified into bottom quartile (low ALK4) and top quartile (high ALK4) groups. Gene set enrichment analysis (GSEA) of TGF-β pathway-related gene signatures was performed, with normalized enrichment scores (NES, $p < 0.05$) indicated by the colored symbols, as defined in the plot legend. **f** Five out of seven TGF-β target genes are significantly negatively correlated with *ACVR1B* in the TCGA breast cancer expression dataset ($n = 960$). **g** Seven TGF-β target genes are significantly and negatively correlated with *ACVR1B* in the TCGA pancreatic cancer expression dataset ($n = 149$). **h–j** MDA-MB-231 CRISPR NTC and ALK4 KO cells, with or without dominant-negative TβRII (DN-TβRII), were serum-starved for 3 h and treated with vehicle or 100 pM TGF-β1 for 30 min. **h** Protein lysates were analyzed by Western blotting. **i** *SERPINE1* expression was measured by qRT-PCR ($n = 3$ biological replicates). **j** Cell migration was assessed using transwell migration assays ($n = 4$ biological replicates). For experiments with two independent variables, statistical analyses were performed using ordinary two-way ANOVA with Tukey's multiple comparisons test. Data are presented as mean values ± SEM.

To investigate how ALK4 loss affects TGF-β receptor glycosylation and stability, we constructed a protein-protein interaction network of the top 125 upregulated DEPs using STRING (Search Tool for the Retrieval of Interacting Genes/Proteins) analysis. The network revealed three distinct clusters (Supplementary Fig. 12f), with the largest cluster predominantly composed of glycoproteins and cell surface receptors (Fig. 6g). Notably, several TGF-β pathway target genes or coreceptors, including FN1, matrix metallopeptidase 2 (MMP2), CD44, and endoglin (ENG), were found to interact with the extracellular scaffolding protein galectin-3 (LGALS3) (Supplementary Data 2). These findings suggest that galectin-3 may play a critical role in the upregulation of TGF-β signaling observed with ALK4 loss.

### ALK4 loss drives cancer progression via galectin-3 and MGAT5-dependent pathways

Previous studies have demonstrated that branching of N-glycans catalyzed by MGAT5 on cell surface receptors promotes their binding to galectin-3, resulting in prolonged receptor activation[18,38]. Consistent with proteomic screen results, we found that ALK4 loss upregulated galectin-3 expression at both mRNA and protein levels in PANC-1 cells (Fig. 7a) and MDA-MB-231 cells (Supplementary Fig. 13a). Next, we specifically evaluated levels of MGAT5-catalyzed β1,6-branch on N-glycans using Phaseolus Vulgaris Leucoagglutinin (PHA-L) lectin in ALK4 KO and control cells. ALK4 loss significantly increased MGAT5-modified glycans in PANC-1 and MDA-MB-231 cells (Fig. 7b, Supplementary Fig. 13b), due to elevated MGAT5 expression (Fig. 7c, Supplementary Fig. 13c) and increased availability of the MGAT5 substrate UDP-GlcNAc (Supplementary Fig. 13d). The increased UDP-GlcNAc was likely due to the increased expression of GFPT2, the rate-limiting enzyme in the hexosamine biosynthesis pathway (HBP) (Supplementary Fig. 13e–g). We next examine whether ALK4 specifically increased MGAT5-catalyzed branched N-glycan. Among the 10 N-branching enzymes tested, only MGAT5 was specifically upregulated in both cell lines following ALK4 loss (Supplementary Fig. 13h, i). An unbiased lectin assay confirmed an increase in β1,6 GlcNAc-branched N-glycans, as detected by PHA-L, while other N-glycan types were not similarly affected (Supplementary Fig. 13j, k).

In ALK4 KO cells, MGAT5-catalyzed glycans on TβRII were significantly increased, as shown by PHA-L lectin pulldown assays (Fig. 7d). Silencing galectin-3 or MGAT5 expression reduced TGF-β-induced Smad2 phosphorylation in ALK4 KO cells (Fig. 7e, f, Supplementary Fig. 14a, b), suggesting that MGAT5 and galectin-3, at least in part, mediate the effects of ALK4 loss. Additionally, MGAT5 knockdown significantly lowered galectin-3 protein levels induced by ALK4 loss (Fig. 7e). Galectin-3 stabilization of TGF-β receptors, which inhibits their internalization, was consistent with the observed reduction in TβRII internalization following ALK4 (Fig. 7g). In a reciprocal manner, restoring ALK4 expression in LM2 cells increased the downregulation of TβRII and TβRI (Fig. 5e).

We next explored the functional consequences of increased MGAT5 expression and activity in the absence of ALK4. Consistent with our previous finding, MGAT5 depletion suppressed the activation of TGF-β signaling (Supplementary Fig. 14c, d). Depletion of MGAT5 significantly inhibited anchorage-independent growth induced by ALK4 loss (Fig. 7h). In vivo, ALK4 loss in PANC-1 cells increased pulmonary lesion formation after tail vein injection into immunodeficient mice (Fig. 7i). MGAT5 KO significantly reduced pulmonary lesion development (Fig. 7i–k). Silencing galectin-3 or MGAT5 also impaired ALK4 KO cell migration while having minimal effects on control cells (Supplementary Fig. 14e, f). Furthermore, knockdown of either galectin-3 or MGAT5 reduced mesenchymal marker expression and TGF-β target gene induced by ALK4 loss (Supplementary Fig. 14g, h). These observations provide new insights into how ALK4 loss drives cancer progression and identify galectin-3 and MGAT5 as potential therapeutic targets in the context of ALK4 loss.

### ETS1 mediates ALK4 loss-induced upregulation of MGAT5

To elucidate how ALK4 loss drives MGAT5 upregulation, we first investigated the involvement of elevated TGF-β signaling. As shown in Supplementary Fig. 15a–i, silencing TβRI or treating cells with the small molecule inhibitor LY2157299 had no significant effect on ALK4 loss-mediated induction of MGAT5 and galectin-3 or MGAT5-modified glycans in PANC-1 and MDA-MB-231 cells. Similarly, SMAD2 knockdown did not alter galectin-3 or MGAT5 expression (Supplementary Fig. 15j, k). We next examined the role of non-canonical TGF-β signaling pathways in MGAT5 regulation in the absence of ALK4. Acute ALK4 inhibition significantly elevated phospho-p38 levels (Supplementary Fig. 16a–c). However, blocking the P38-MAPK pathway with the small molecule inhibitor SB203580 did not affect TGF-β signaling or MGAT5 expression (Supplementary Fig. 16d–f).

Previous studies identified ETS1, ETS2, and JUN as direct regulators of MGAT5[39–41]. Among these, only ETS1 showed a strong negative correlation with ACVR1B and a positive correlation with MGAT5 in TCGA pancreatic and breast cancer cohorts (Fig. 8a–c), implicating ETS1 in ALK4 loss-mediated MGAT5 upregulation. Indeed, in ALK4 KO cells, ETS1 expression was significantly elevated, and ETS1 knockdown significantly reduced MGAT5 levels induced by ALK4 loss (Fig. 8d–f). These findings establish ETS1 as a key mediator of MGAT5 upregulation following ALK4 loss. Consistent with our previous findings, we observed a significant increase in the expression of Ets1, Galectin-3, and Mgat5 in ALK4-deficient KC mice (Supplementary Fig. 17a–c). We also observed increased expression of several TGF-β-regulated genes that are involved in EMT in the pancreas from AKC mice compared to the pancreas from KC mice (Supplementary Fig. 17d–h). These findings further support our model that loss of ALK4-activated TGF-β signaling results from regulating MGAT5 expression.

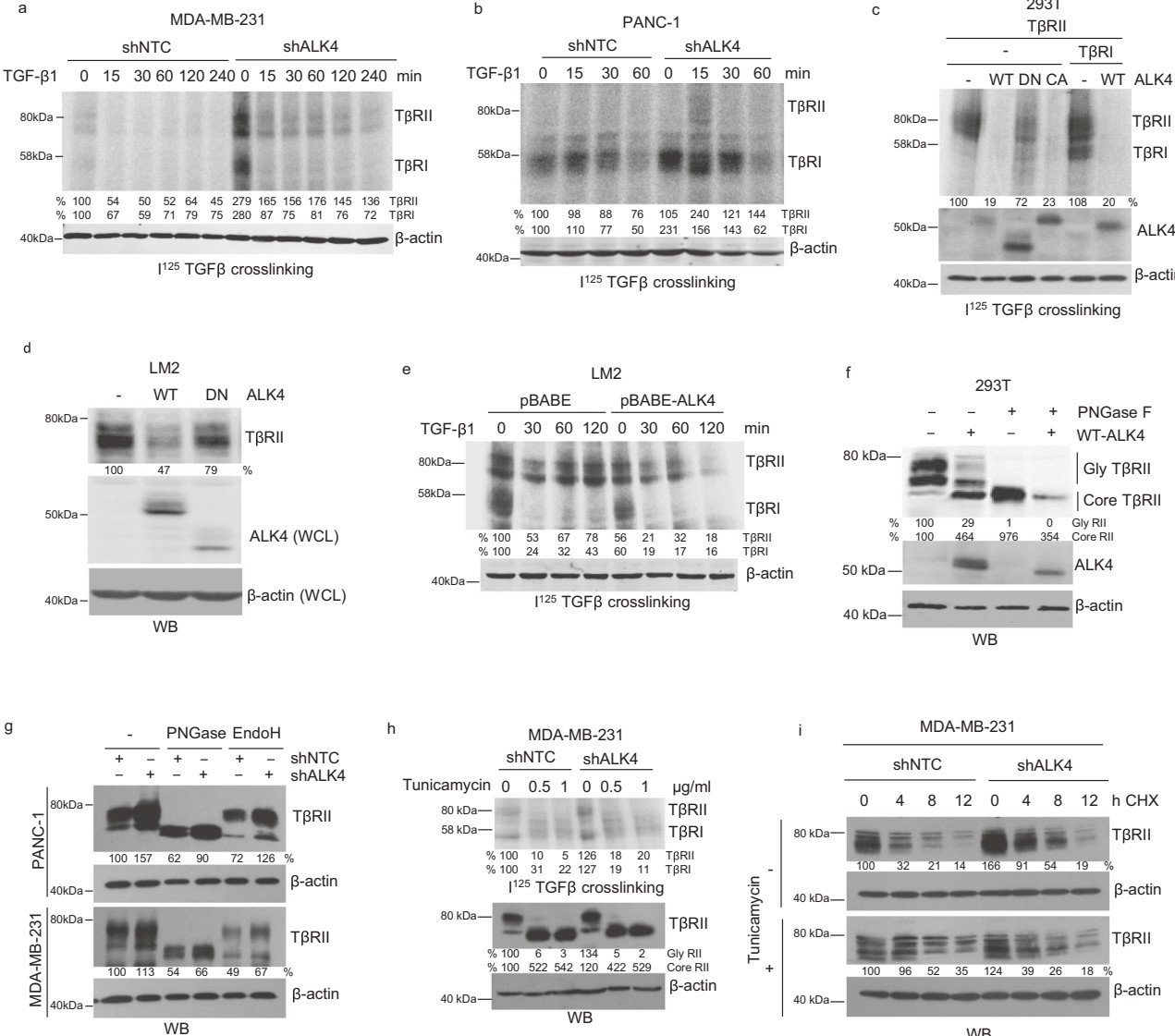

**Fig. 5 | ALK4 loss increases cell surface levels of TGF-β receptors. a–c** Cell surface levels of TGF-β receptors were assessed using I[125]-TGF-β binding and crosslinking assays. MDA-MB-231 (**a**) and PANC-1 (**b**) cells expressing shNTC control or shALK4 were serum-starved for 3 h and treated with 100 pM TGF-β1 for the indicated times. **c** 293T cells were transfected with expression plasmids for TβRII and/or TβRI, along with wild-type (WT), constitutively active (CA), or dominant-negative (DN) ALK4. ALK4 and β-actin were analyzed. **d** LM2 cells were transfected with expression plasmids for TβRII, along with wild-type (WT) or dominant-negative (DN) ALK4. Biotinylated cell surface proteins were analyzed for TβRII, and whole cell lysates were analyzed for ALK4 and β-actin using Western blotting. **e** LM2 cells expressing pBABE control or pBABE-ALK4 were serum-starved for 3 h and treated with 100 pM TGF-β1 for the indicated times. Cell surface levels of TGF-β receptors were assessed using I[125]-TGF-β binding and crosslinking assays.

**f** 293T cells were transfected with WT ALK4 as indicated, and total cell lysates were processed with PNGase F and assessed by Western blotting. **g** PANC-1 and MDA-MB-231 cells expressing shNTC or shALK4 were lysed. Total cell lysates were processed with PNGase F or EndoH and assessed by Western blotting. **h** MDA-MB-231 cells stably expressing shNTC or shALK4 were treated with tunicamycin at the indicated concentrations. Cell surface levels of TGF-β receptors were assessed using an I[125]-TGF-β binding and crosslinking assay (top). Total TβRII and β-actin were assessed by Western blotting. **i** MDA-MB-231 cells expressing shNTC or shALK4 were treated with 10 nM cycloheximide (CHX) for the indicated times to block synthesis of new proteins. Cells were incubated with 50 ng/ml tunicamycin for 4 h prior to harvesting. Whole-cell lysates were assessed for TβRII and β-actin by Western blotting. Quantification was performed using ImageJ. For in vitro analysis, each experiment was done with at least three independent biological replicates.

## Inhibiting N-linked glycosylation effectively suppresses TGF-β signaling and tumor metastasis driven by ALK4 loss

We investigated whether inhibiting ALK4 loss-induced N-glycosylation could mitigate cancer progression. Two N-glycosylation inhibitors, N-linked glycosylation inhibitor 1 (NGI-1)[42] and glucosamine[43,44], were tested. Both inhibitors reduced MGAT5-catalyzed glycans, as detected by PHA-L staining (Supplementary Fig. 18a, b), to levels similar to those observed in MGAT5 CRISPR KO cells (Supplementary Fig. 14c). NGI-1 reduced TGF-β-induced signaling in both control and ALK4 KO cells, while glucosamine exhibited a stronger suppressive effect, particularly in ALK4 KO cells (Supplementary Fig. 18c, d). NGI-1 treatment also

significantly decreased cell invasion in ALK4 KO cells (Supplementary Fig. 18e, f). In addition, both NGI-1 and glucosamine suppressed anchorage-independent growth in control and ALK4 KO cells, whereas the TβRI inhibitor LY2157299 had more modest effects in vitro (Supplementary Fig. 18g, h). Importantly, these effects were not attributable to changes in the number of viable cells (Supplementary Fig. 18i).

A major consequence of ALK4 loss is the promotion of cancer progression. To determine whether inhibiting N-glycosylation could counteract this effect, we assessed pulmonary lesion formation, using LY2157299 as a positive control. In line with the in vitro results and prior data, ALK4 loss significantly increased pulmonary lesions.

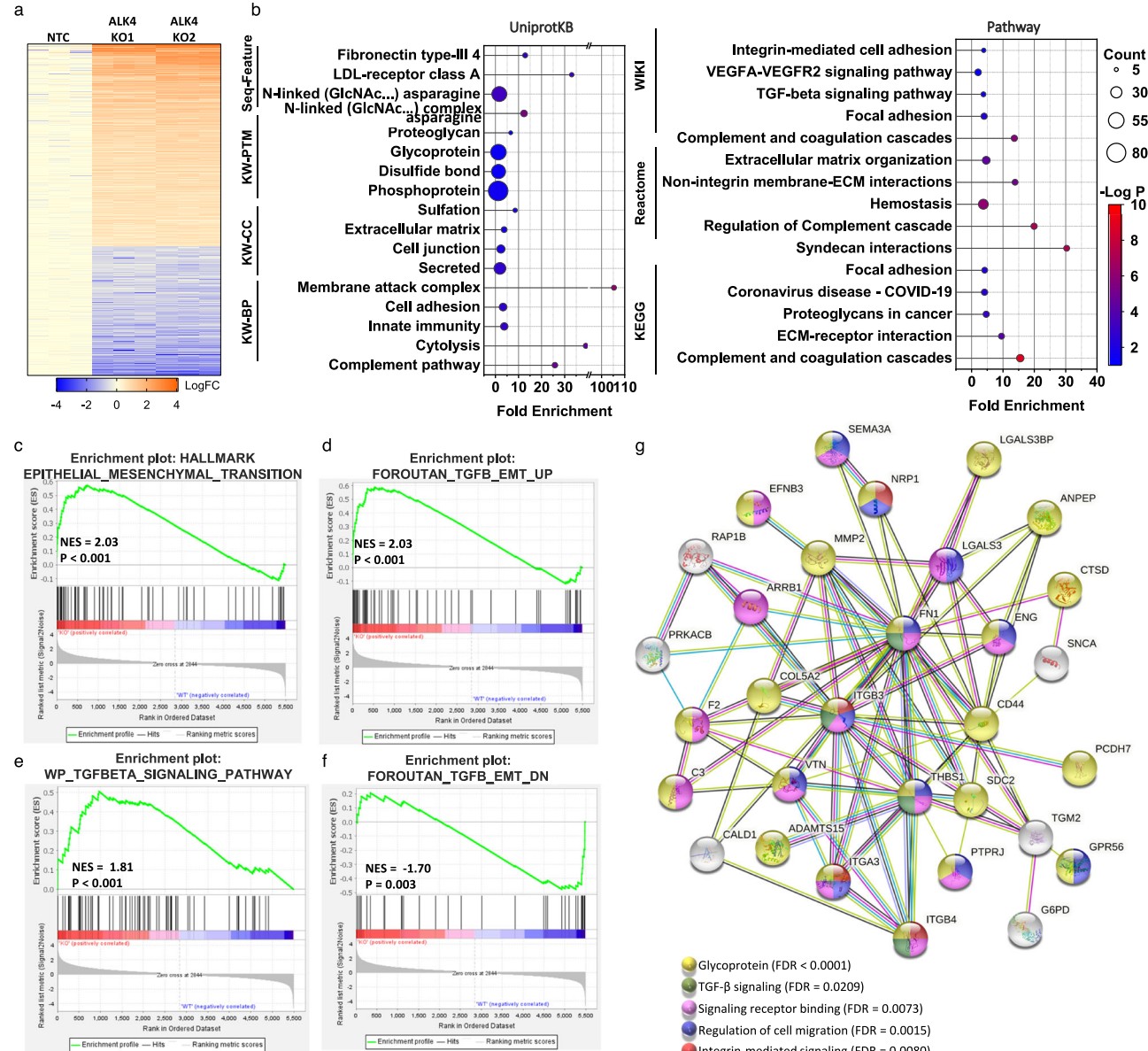

**Fig. 6 | Proteomic analysis reveals potential mechanisms by which ALK4 loss promotes TGF-β signaling and cancer progression. a** Quantitative proteomic analysis of whole-cell lysates from PANC-1 control (NTC) and two ALK4 CRISPR knockout (ALK4 KO) lines. Heatmap showing the expression of differentially expressed proteins (DEPs, *p* < 0.05) between ALK4 KO and control samples. Colors represent relative expression values in log fold change (logFC). **b** Functional annotation and pathway enrichment analysis of the top 125 upregulated DEPs in ALK4 KO samples compared to controls using DAVID Bioinformatics. Significant gene sets are shown with circle size indicating the number of genes and color indicating the transformed p-value. Fold enrichment is represented on the *x*-axis. **c–f** Gene set enrichment analysis (GSEA) plots for DEPs (KO vs. WT) showing significant enrichment in indicated gene signatures. NES: normalized enrichment score. **g** Protein–protein interaction network of the top-regulated proteins in ALK4 KO cells, generated using STRING analysis. Interactions with a score > 0.5 are depicted, with line color indicating the type of interaction (cyan-curated databases; pink-experimentally determined; blue-gene co-occurrence; khaki-text mining; black-co-expression; light blue-protein homology). Dots represent the pathways/biology as indicated.

Treatment with NGI-1 or LY2157299 significantly reduced lesion formation, underscoring the potential of N-glycosylation inhibition to suppress cancer progression (Supplementary Fig. 19a–d). In mice injected with ALK4 KO cells, we observed more cell surface PHA-L stain in the neoplastic region, and this signal was reduced by treatment with NGI-1 (Supplementary Fig. 19e).

These findings support a model in which ALK4 loss promotes MGAT5-mediated N-glycosylation of cell surface proteins, including canonical TGF-β receptors. This modification stabilizes the receptors at the cell surface by facilitating galectin-3 binding, thereby reducing receptor internalization. The resulting elevated receptor levels enhance TGF-β receptor complex formation, driving EMT, migration,

invasion, and cancer progression. ETS1 at least partially mediates the upregulation of MGAT5 following ALK4 loss. Importantly, N-glycosylation inhibitors effectively suppress ALK4 loss-induced tumor progression.

## Discussion

The TGF-β signaling pathway plays a well-established dichotomous role in human cancers. Loss of TGF-β signaling is associated with tumor initiation, while hyperactivation of this pathway promotes cancer progression, angiogenesis, metastasis, and immune evasion[8,45,46]. Given this role in cancer progression, TGF-β signaling has remained an attractive target for cancer therapy. Over the years, a number of

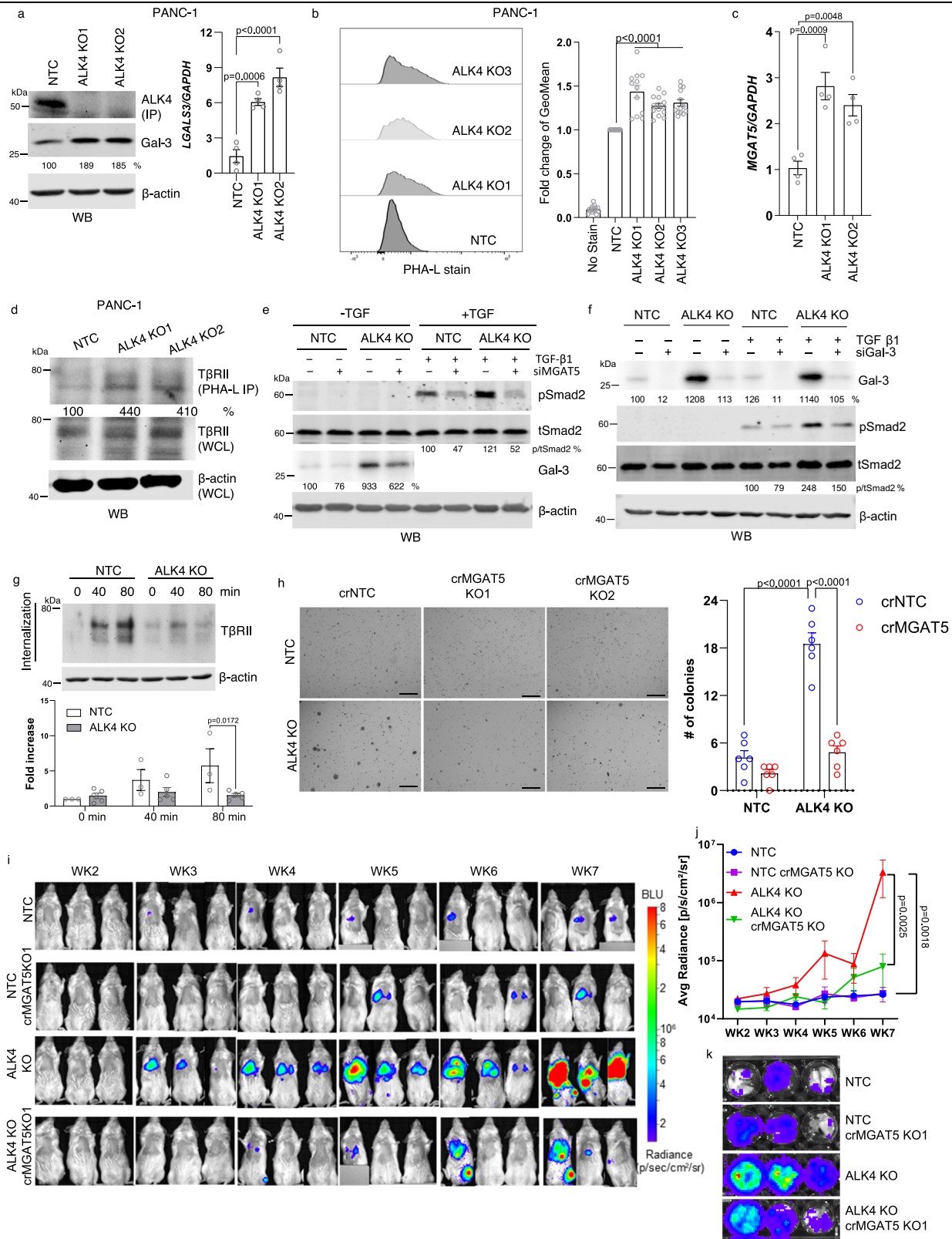

inhibitors of TGF-β signaling have been developed, including monoclonal antibodies against ligands, latent TGF-β stabilizing mAbs, decoy receptors, and TGF-β receptor kinase inhibitors[6,14,47]. These approaches have shown promise in preclinical models by suppressing TGF-β signaling and inhibiting tumor progression[48–50]. Despite these advances, clinical development has been hindered by adverse effects, notably cardiac toxicity[51,52], and a lack of reliable biomarkers for patient selection[14,53]. Given these challenges, there is a critical need to identify

novel regulators and molecular mechanisms that regulate TGF-β signaling, particularly those offering more effective and tolerable therapeutic strategies.

In this study, we reveal an important role of the tumor suppressor ALK4 in suppressing TGF-β signaling and cancer progression. Specifically, we show that loss of ALK4 induces aberrant N-glycosylation, which stabilizes TGF-β receptors and enhances signaling. Targeting N-glycosylation in ALK4-deficient pancreatic and triple-

**Fig. 7 | Loss of ALK4 enhances cancer progression through upregulating galectin-3 and MGAT5-driven glycosylation. a** Western blot (left) and qRT-PCR (right) analysis of protein and *LGALS3* mRNA expression in PANC-1 NTC and ALK4 KO cells (*n* = 4 per group). **b** Flow cytometric analysis of MGAT5-modified glycans labeled with PHA-L lectin in PANC-1 NTC (*n* = 14), ALK4 KO1 (*n* = 12), ALK4 KO2 (*N* = 14), and ALK4 KO3 (*n* = 13) cells. **c** qRT-PCR analysis of *MGAT5* expression in PANC-1 NTC and ALK4 KO cells (*n* = 4 per group). **d** Magnetic beads were coated with PHA-L lectin and incubated with whole-cell lysates from PANC-1 NTC or ALK4 KO cells. Eluted samples were analyzed for TβRII. Whole-cell lysates (WCL) were analyzed for TβRII and β-actin. **e** PANC-1 control and ALK4 KO cells transfected with control or MGAT5-specific siRNA. After 2 days, cells were transfected again for an additional 3 days, serum-starved for 3 h, and treated with 100 pM TGF-β1 for 30 min before harvesting. Protein lysates were assessed for expression of the indicated proteins using western blotting. **f** Western blot analysis of protein expression in PANC-1 NTC and ALK4 KO cells transfected with control or galectin-3-specific siRNA for 5 days. Cells were treated with 100 pM TGF-β1 for 30 min before harvesting. **g** Internalization of TβRII in PANC-1 NTC (*n* = 3) and ALK4 KO cells (*n* = 5). Quantification of internalized TβRII under the indicated condition is shown at the bottom. **h** Resullts of soft agar colony formation assay of indicated groups (*n* = 6 per group), scale bar = 200 nm. **i–k** In vivo pulmonary lesions in NSG mice injected via tail vein with PANC-1 NTC, PANC-1 NTC crMGAT5 KO, ALK4 KO, or ALK4 KO crMGAT5 KO cells (*n* = 10 per group). Pulmonary lesions were imaged via bioluminescent imaging in a blinded manner (**i** and **j**). **k** Bioluminescent imaging of lungs from the top three mice (highest bioluminescence) from each group at week 7. For in vitro analysis, each experiment was done with at least 3 independent biological replicates. For multiple comparisons with one independent variable, ordinary one-way ANOVA was used, followed by Dunnett's multiple comparisons test. For experiments with two independent variables, ordinary 2-way ANOVA was used, followed by Tukey's multiple comparisons test. Data are presented as mean values ± SEM.

negative breast cancer models not only suppresses TGF-β signaling but also inhibits cancer progression. These findings suggest a promising therapeutic strategy for targeting TGF-β-driven cancer progression and highlight ALK4 loss and/or N-glycosylation as potential biomarkers to identify patients who may benefit from such treatments. Although the cancer progression phenotypes induced by ALK4 loss were consistent across breast and pancreatic cancer models, the in vivo and mechanistic validation is more extensive in pancreatic cancer models. Further studies, including validation of the role of ALK4 loss in an autochthonous mouse model of breast cancer, would provide stronger support for the proposed mechanism in the context of breast cancer.

ALK4 has previously been identified as a tumor suppressor[22,54]. However, ALK4 expression and its prognostic potential have not been well investigated. Here, we investigated in silico patient data and clinical specimens to determine that ALK4 is decreased in pancreatic and breast cancer patients, as well as in other cancers. Besides somatic mutation, somatic biallelic inactivation of the *ACVR1B* locus, and loss of heterozygosity as previously reported[22,23], lower ALK4 expression in tumors also results from loss of copy number and epigenetic regulation, including promoter hypermethylation as demonstrated here.

ALK4 mediates activin A signaling via Smad2/3 activation[55]. Similar to TGF-β, activin A signaling exhibits a dual role during tumorigenesis, promoting anti-proliferative and tumor-suppressive effects[56,57] while also inducing EMT and enhancing tumor progression in breast and pancreatic cancers under specific conditions[57–60]. In this study, we found that inhibiting activin A signaling with sActrIIB-Fc did not replicate the effects of ALK4 loss on TGF-β signaling or EMT. Instead, activin A inhibition reduced mesenchymal marker expression and increased E-cadherin levels. These findings suggest that the loss of ALK4 function influences cancer progression through mechanisms beyond activin A signaling. Indeed, a recent study demonstrated that inhibiting activin A signaling or knocking out ALK4 in KRAS[G12D]-expressing mouse models reduced senescence and increased proliferation during the early stages of pancreatic tumorigenesis[26]. These observations underscore the multifaceted role of ALK4 loss, which likely drives cancer progression through disrupted activin A signaling, enhanced TGF-β signaling, and additional pathways regulated by the galectin-3/MGAT5 axis.

Given the similarities between ALK4 and ALK5, the type I TGF-β receptor in terms of structure, evolution, and function[61,62], we considered the possibility that ALK4 might simply compete with ALK5 for a common factor regulating their cell surface stability. However, the effects of ALK4 loss were not restricted to ALK5 or to homologous receptors, extending to other cell surface receptors, including the integrins and VEGFR2 receptor, leading to our investigation of a broader mechanism for regulating cell surface expression, through glycosylation and retention at the cell surface.

MGAT5 is overexpressed in breast and pancreatic cancer[63,64]. Increased MGAT5 expression in cancer promotes EMT and tumor invasiveness and is strongly associated with a poor prognosis[65–67]. While there have been several mechanisms that have been reported to contribute to the regulation of MGAT5 expression, including transcription factors, non-coding RNAs, or regulators of MGAT5 mRNA stability[68–70], here we have demonstrated that loss of ALK4 mediated increases in the ETS1 transcription factor, which may account for increased MGAT5 expression. How ALK4 loss of function leads to increased levels of ETS1 at the message level remains to be explored. Future unbiased omics-based analyses will be valuable for deepening our understanding of ALK4-dependent transcriptional programs.

MGAT5-catalyzed glycans on kinase receptors are preferentially recognized by galectin-3, which binds to these modified glycans to form ordered clusters on the cell surface. These clusters prevent receptor endocytosis, stabilizing glycoproteins and enhancing their signaling potential[18,71]. The MGAT5-galectin-3 interaction serves as a general stabilizer of cell surface glycoproteins[72–74]. Galectin-3 interacts with multiple cell surface receptors, including EGFR and integrins[18,75], regulates signaling pathways, including VEGFR2 signaling[76], and stabilizes glycoproteins, including fibronectin and N-cadherin[71]. Our study highlights the upregulation of galectin-3 as a significant effect of ALK4 loss in cancer cells, suggesting that the loss of ALK4 function alters the glycosylation and function of a number of cell surface and secreted proteins. Proteomic analysis revealed that ALK4 loss triggered the upregulation of several glycoproteins and cell surface receptors, including integrin β4, integrin β3, and endoglin. These findings indicate a broader regulatory role for MGAT5/galectin-3-mediated stabilization in ALK4-deficient cancers. Ongoing investigations are exploring how MGAT5 and galectin-3 modulate other key receptors, including VEGFR and integrins, in cancer models, which could provide additional therapeutic strategies.

## Methods

All animal experiments were conducted under protocols approved by the Duke University Institutional Animal Care and Use Committee (Protocol Number A130-23-06). In the tumorigenicity study, the tumor size didn't exceed the maximal tumor burden allowable (2000 mm³). All recombinant DNA experiments were conducted under protocols approved by the Duke Institutional Biosafety Committee.

### Cell lines and reagents

The human breast cancer cell line MDA-MB-231 was obtained from the American Type Culture Collection (ATCC). LM2 cells, a derivative of MDA-MB-231, were provided by Dr. Joan Massagué (Memorial Sloan Kettering Cancer Institute). Both MDA-MB-231 and LM2 cells were cultured in Dulbecco's modified Eagle medium (DMEM, Thermo Fisher Scientific) supplemented with 10% fetal bovine serum (FBS)

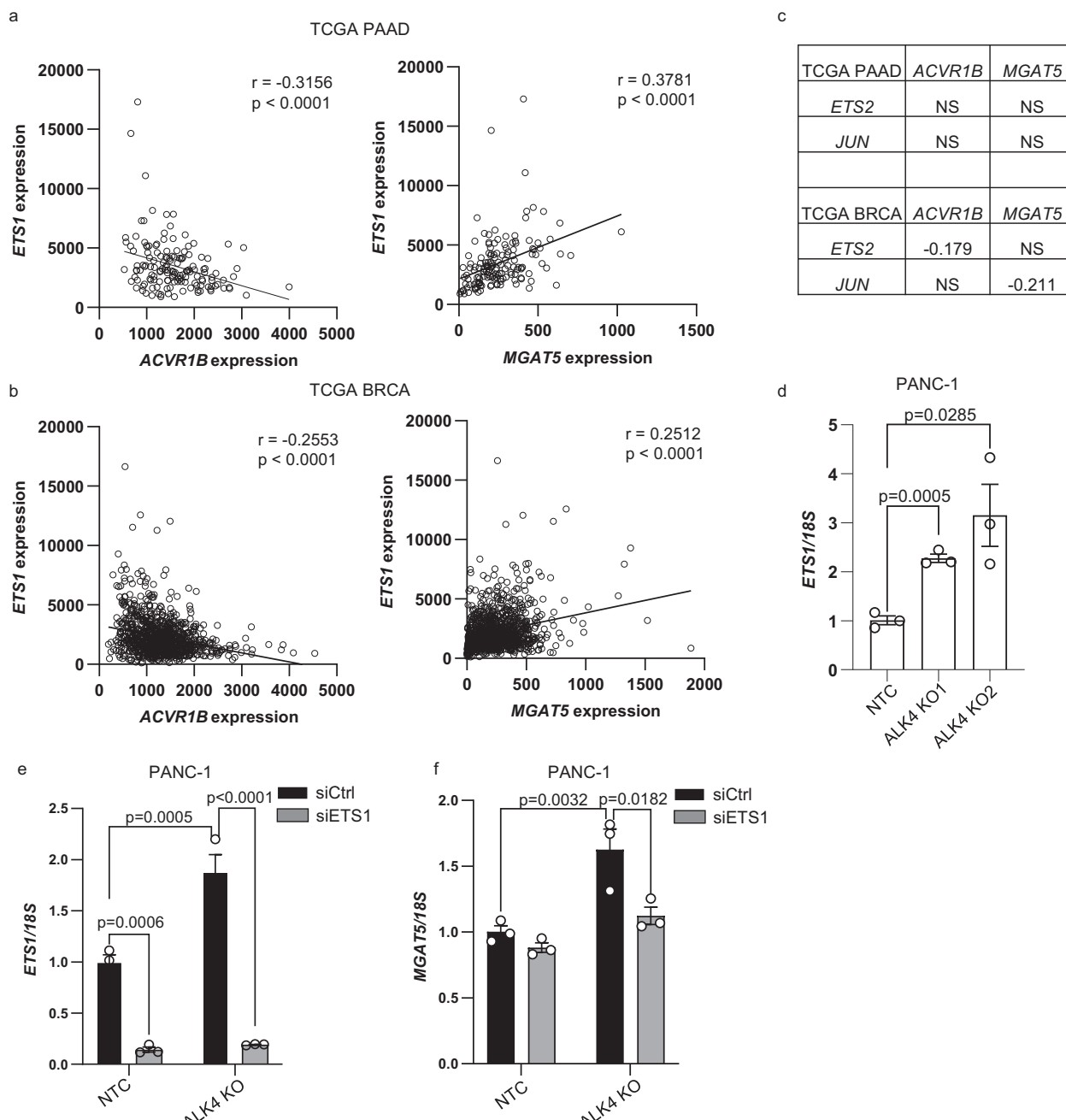

**Fig. 8 | Loss of ALK4 promotes MGAT5 expression through the induction of ETS1.** Dot plots of the correlation between *ACVR1B* and *ETS1* expression (left) and between *MGAT5* and *ETS1* expression (right) using the TCGA pancreatic (**a**) and breast (**b**) cancer databases. **c** Spearman correlation coefficients between the indicated genes in the TCGA pancreatic and breast cancer databases. NS not significant. **d** qRT-PCR analysis of *ETS1* mRNA levels in PANC-1 NTC and ALK4 KO cells. qRT-PCR analysis of *ETS1* (**e**) and *MGAT5* (**f**) PANC-1 NTC and ALK4 KO cells transfected with control or ETS1-specific siRNA for 48 h. Results were repeated with three independent biological replicates. For experiments with two independent variables, data were analyzed using ordinary two-way ANOVA followed by Tukey's multiple comparisons test. For comparison between two groups, data were analyzed using two-tailed Student's *t* tests. Data are presented as mean values ± SEM.

(Corning) and 1% penicillin/streptomycin (Thermo Fisher Scientific). The MCF-10A human breast cancer progression series of cell lines were provided by Dr. William Schiemann (Case Western Reserve University) and maintained in DMEM/F12 medium (Thermo Fisher Scientific) supplemented with 5% horse serum (Corning), 10 μg/ml insulin (Thermo Fisher Scientific), 20 ng/ml epidermal growth factor (Sigma), 100 ng/ml hydrocortisone (Sigma), 100 ng/ml cholera toxin (Sigma), and 1% penicillin/streptomycin. The 4T1 mouse breast cancer cell line series was also provided by Dr. William Schiemann and

maintained in DMEM supplemented with 10% FBS. The human pancreatic cell lines PANC-1 and HPNE (ATCC) were maintained in DMEM medium supplemented with 10% FBS and 1% penicillin/streptomycin. The ALK4 WT (Cat# 80879), ALK4 CA (Cat# 27223), TβRII DN (Cat# 12640), TβRI (Cat# 14831), and TβRII (Cat# 11766) DNA plasmids were purchased from Addgene. The ALK4 DN DNA plasmid[77] was provided by Dr. Anne Klibanski. The inhibitors SB431542, LY2157299, SB203580, and NGI-1 were purchased from Selleckchem. Glucosamine was purchased from Sigma.

## Patient sample analysis

mRNA from pancreatic adjacent healthy tissue ($n = 13$) and primary pancreatic tumor tissue ($n = 20$) was acquired from the Duke BioRepository & Precision Pathology Center (Supplementary Data 3), with informed patient consent and approved by the Duke University IRB (deidentified, exempt). All samples were reviewed and approved by board-certified pathologists. Quantitative reverse transcription polymerase chain reactions (qRT-PCR) were performed using primers specific for *ACVR1B* and *GAPDH* (Supplementary Data 1). *ACVR1B* mRNA expression was normalized to *GAPDH* using the ΔΔCt method. Statistical significance was assessed using the Mann–Whitney test, assuming a non-Gaussian distribution for comparisons between two groups.

## Generation of stable cell populations

PT67 packaging cells (Clontech) were transfected with shNTC, shALK4, or shALK4-rALK4 constructs to produce retroviral stock, which was subsequently used to infect HPNE, PANC-1, MD-MB-231, and MCF10A, following previously described protocols[78]. LM2 cells were transfected with pBABE or pBABE-ALK4 using Lipofectamine 3000 (Thermo Fisher Scientific), and stable cell populations were selected as recommended[79].

CRISPR-mediated *ALK4* knockout cells were generated using the pLentiCRISPRv2-puro vector (Addgene, Cat# 98290). Three gRNA sequences were annealed separately into BsmBI-digested pLentiCRISPRv2-puro vector using T4 ligase, and gRNA insertion was confirmed by Sanger sequencing. The finalized pLentiCRISPRv2-puro-gRNA vector was co-transfected into 293T cells along with psPAX2 (Addgene) and pMD2.G (Addgene) plasmids. Lentiviral-conditioned medium, supplemented with polybrene (Sigma, Cat# TR-1003-G), was used to infect parental Panc1 or MDA-MB-231 cells, followed by puromycin selection (1 μg/ml). All newly generated materials (stable cell lines) are available upon request. The shRNA and gRNA sequences used in this study are listed in Supplementary Data 1.

## In vivo tumorigenicity and metastasis model

All mice used in this study are housed in IVCs (Individually ventilated cages) under positive air pressure in a barrier facility. The light cycle is 12 on and 12 off; typically, lights are on from 7 a.m. to 7 p.m. Temperature is $72\,°C \pm 2$ on either side, and humidity can range from 30% to 70%.

HPNE cells ($5 \times 10^6$) stably expressing shNTC or shALK4 in RPMI medium with 10% Matrigel were orthotopically injected into the pancreata of 6-week-old male nude mice (NU/J, JAX:002019). After 15 weeks, mice were euthanized for analysis of primary tumors and metastases. All metastatic lesions noted were detected in different mice (one metastatic lesion per mouse). 4T1-luc cells stably expressing pBABE or pBABE-ALK4 ($10^6$ in 50 μl DPBS) were orthotopically injected into the fourth mammary pad of 6–8-week-old male C57BL/6J mice (JAX:000664). Tumor growth was monitored every other day until day 18. Primary tumors were removed, and metastases were assessed on days 10 and 20 post-resection.

For tail vein injection models, MDA-MB-231-luc cells stably expressing shNTC, shALK4, or shALK4 + rALK4 ($10^6$ cells in 100 μl PBS), LM2-luc cells expressing pBABE or pBABE-ALK4 ($5 \times 10^5$ cells in 100 μl PBS) were injected into female nude mice (JAX:005557, 6–8-week-old). PANC-1-Luc cells stably expressing crNTC or crALK4, and their corresponding MGAT5 KO cells ($5 \times 10^5$ cells in 50 μl PBS) were injected into NSG mice (JAX:005557, 6–8-week-old). An equal number of male and female mice were used. Bioluminescence imaging was performed at specified time points after intraperitoneal injection of D-luciferin (150 mg/kg) using a Caliper IVIS Spectrum in a blinded manner. For drug testing, PANC-1-luc-crNTC or PANC-1-luc-crALK4KO cells ($5 \times 10^5$ cells in 50 μl DPBS) were injected via tail vein into NSG mice. NGI-1 (20 mg/kg, i.p., every other day) and LY2157299 (75 mg/kg, oral, daily)

were prepared in 10% DMSO, 40% PEG300, 5% Tween80, and 45% PBS. IVIS imaging began 5 weeks post-injection.

In the genetically engineered mouse model, *ALK4^flox/flox^* mice were provided by Dr. Philippe Bertolino (Cancer Research Centre of Lyon), *LSL-KRAS^G12D/+^* mice were provided by Dr. David Kirsh from Duke University, and *Pdx1-cre* mice (JAX:014647) were purchased from Jackson Laboratory. *LSL-Kras^G12D/+^* mice were crossed with *Pdx1-Cre* mice to generate the *LSL-KRAS^G12D/+^;Pdx-cre* (KC) mice. *LSL-KRAS^G12D/+^;Pdx-cre* mice were crossed with *ALK4^flox/flox^* mice to generate the *ALK4^flox/flox^;LSL-KRAS^G12D/+^;Pdx-cre* (AKC) mice. KC and AKC were then followed until 8–9 months of age or until humane endpoints were reached. The pancreatic sections and liver sections were evaluated by a board-certified veterinary pathologist in a blinded manner (Table 2).

## siRNA transfection

Silencer Select siRNA targeting ALK4 (S978 and S979), LGALS3 (S8150 and S8149), MGAT5 (S8738 and S8739), TGFBR1 (S14071 and S14073), ETS1 (S4847), SMAD2 (S8397), and a noncoding control (Thermo Fisher Scientific) were transfected at 10–30 nM using Lipofectamine RNAiMAX Transfection Reagent (Thermo Fisher Scientific), following the manufacturer's instructions. For transfection experiments exceeding 3 days, cells were re-transfected every 2–3 days and incubated for an additional 36–60 h before subsequent experiments.

## Transwell migration and invasion assays

Transwell migration was assessed by seeding $1.5–2.5 \times 10^4$ cells in serum-free medium into the upper chamber of a transwell insert with 8.0 μM pores (Corning). Cells were allowed to migrate overnight at 37 °C toward a complete growth medium containing 10% serum in the lower chamber. For transwell invasion assays, $1–5 \times 10^5$ cells were seeded into the upper chamber of Matrigel-coated transwell inserts (Corning) with 8.0 μM pores. Invasion was assessed following incubation at 37 °C for 14–24 h. Migrated or invaded cells were counted in a blinded manner by multiple lab members using ImageJ.

## Cell proliferation assay

The ³H thymidine incorporation assay was performed as previously described[80]. Briefly, cells were incubated with 10 μCi of ³H thymidine (Amersham Biosciences) for 4 h. After gentle washing with 1× PBS and 5% trichloroacetic acid, cells were collected in 0.1 N NaOH. The amount of incorporated ³H thymidine was determined by scintillation counting.

## 3D morphogenesis assay

The three-dimensional culture of cells was carried out as described previously[81]. Briefly, single-cell suspensions of 5000 cells were seeded per well on solidified Matrigel (Corning, Cat# 356231) in the corresponding assay medium. The assay medium was replaced every 48 h. Images of the 3D structures were taken on Day 8. For LM2 cells, the 3D structures were stained with Phalloidin and DAPI.

## Soft agar assay

Soft agar assays were performed in six-well plates (VWR). Each well contained a bottom layer of 0.5% agarose (VWR), a middle layer of 0.3% agarose containing $5 \times 10^3$ cells, and a top layer of DMEM media supplemented with 10% FBS. The top layer of medium was changed every 6 days. After 24 days of growth at 37 °C/5% $CO_2$, colonies were stained with a 0.005% crystal violet solution (Sigma). Colonies with a diameter >200 μm were counted using ImageJ in a blinded manner by multiple lab members.

## Western blot

Preparation of cellular lysates and immunoblotting was performed as previously described[82]. SDS–PAGE was performed using gels ranging from 8% to 12% polyacrylamide that were loaded with 10–30 μg of

protein, as calculated by the Bradford assay, following the manufacturer's instructions (Bio-Rad). Proteins were transferred to 0.45 μm PVDF membranes (EMD Millipore) using a Trans-Blot Turbo Transfer System (Bio-Rad). Membranes were blocked in 5% bovine serum albumin (BSA, NEB), followed by incubation with primary antibodies at 4 °C for 18 h. Membranes were then washed with TBS-Triton (pH 7.4, 10% Triton-X) and incubated with fluorophore-labeled or HRP-conjugated secondary antibodies. Fluorophore-labeled antibodies were developed using an Odyssey® DLx Imaging System (LI-COR). HRP-conjugated antibodies were incubated for 2 min in SuperSignal™ West Pico PLUS Chemiluminescent Substrate (Thermo Fisher Scientific) before imaging with a Bio-Rad ChemiDoc XRS+ Imaging System (Bio-Rad). Antibodies used are listed in Supplementary Data 1. Western blot qualification was performed using Image Studio or ImageJ.

## Cell surface biotinylation and co-immunoprecipitation
For ALK4 knockdown/knockout validation, $6 \times 10^6$ cells were plated in 10-cm plates and incubated with NHS-Biotin (0.5 mg/ml in ice-cold PBS) for 30 min in the dark before being collected. ALK4 protein was pulled down using an ALK4 antibody (R&D) and detected using HRP-conjugated streptavidin (Thermo Fisher Scientific).

Cell surface biotinylation experiments were performed using a Cell Surface Biotinylation and Isolation Kit (Thermo Fisher Scientific) following the manufacturer's instructions. Briefly, control or ALK4 KOs ($12 \times 10^6$ cells/15-cm plates) were washed twice with ice-cold PBS. Cell surface biotinylation was conducted by adding 10 ml of sulfo-NHS-SS-biotin solution (0.25 mg/ml in ice-cold PBS) to each plate. The biotin labeling reaction was performed by incubation at 4 °C and quenched after 30 min by washing twice with 20 ml of ice-cold TBS. Cells were collected in TBS, and cell pellets were lysed. The biotinylated proteins were captured with Pierce™ NeutrAvidin™ Agarose (Thermo Fisher Scientific) and eluted in elution buffer supplemented with dithiothreitol (DTT). Samples were stored at −80 °C for future analysis.

## Cell surface receptor internalization
Cells were washed with ice-cold PBS containing 1 mM $MgCl$ and 0.1 mM $CaCl_2$ (PBS++) and incubated for 30 min with 0.5 mg/ml EZ-Link Sulfo-NHS-SS-Biotin (Thermo Scientific) in PBS++ on ice. Cells were then incubated in complete growth medium in a 37 °C incubator for the indicated times. Internalization was halted by transferring cells back to ice and washing them with ice-cold PBS. Remaining cell surface biotin was removed by washing three times with a mildly reducing buffer containing 50 mM glutathione, 75 mM NaCl, 75 mM NaOH, and 10% FBS in PBS++ for 15 min. Samples were then harvested and lysed, and biotinylated receptors were collected as described in the Cell surface biotinylation and co-immunoprecipitation section.

## Immunofluorescence assay
Formaldehyde-fixed cells were permeabilized with 0.2% Triton X-100, blocked with 8% BSA, and stained with the appropriate antibodies, including anti-β-catenin (1:200), anti-ZO-1 antibody (1:200), anti-vimentin antibody (1:200), anti-fibronectin (1:200), and anti-Smad2 (1:100). Cells were then incubated with Alexa Fluor-488-conjugated anti-mouse IgG (1:200) or Alexa Fluor-488-conjugated anti-rabbit IgG (1:200). Images were acquired using a Leica SP5 inverted confocal microscope or an Olympus FV1000 confocal laser scanning microscope.

## Lectin arrays
A Lectin Array 70 kit was purchased from RayBiotech (GA-Lectin-70). Assays were performed as per the manufacturer's instructions using a label-based approach. The whole cell lysates from NTC and ALK4 KO cells were used in this assay. The intensity was detected and analyzed using a SensoSpot Fluorescence Microarray Analyzer.

## UDP-GlcNAc assay
The UDP-GlcNAc assay was developed by Duke Proteomics and Metabolomics Core Facility. Briefly, PANC-1 crNTC and crALK4 KO cells and MDA-MB-231 crNTC and crALK4 KO cells (4 replicates for each condition) were lysed and dried down under nitrogen at room temperature. Samples were incubated with $^{13}C_6$-UDP-GlcNAc (Omicron Biochemicals, Inc.) in a 20% acetonitrile solution. Samples were analyzed with a multiple reaction monitoring (MRM) assay. Data was acquired using UPLC-MS/MS on a Sciex QTrap 6500+ mass spectrometer, including calibration curves for the analyte in a manner consistent with the FDA Guidance for Bioanalytical Method Validation. Data analysis was performed using Skyline.

## Bioinformatic data analysis
RNA sequencing of level 3 data from human breast and pancreatic adenocarcinoma tumors was obtained from The Cancer Genome Atlas (TCGA)[83–85] through the Firehose data portal (https://gdac.broadinstitute.org/). RNA-Seq by Expectation-Maximization values were log-transformed and median-centered. Genes present in fewer than 20% of patients were excluded. Illumina HT-29 v3 expression data for the Molecular Taxonomy of Breast Cancer International Consortium (METABRIC) project were obtained from the European Genome-phenome Archive (https://www.ebi.ac.uk/ega/) and median-centered[86]. Breast cancer subtypes were classified using the Prediction Analysis of Microarray (PAM) 50 classifier (PAM50)[87].

Gene expression data from pancreatic cancer patients from Clinical Proteomic Tumor Analysis Consortium (CPTAC) and Queensland Clinical Molecular Genetics (QCMG) datasets were downloaded from cBioPortal (https://www.cbioportal.org/datasets).

To examine *ACVR1B* expression in breast cancer subtypes, gene expression data were z-scored and plotted according to PAM50 classifications. Statistical significance between subtypes was assessed using a two-way analysis of variance (ANOVA). For specific gene analysis relative to *ACVR1B* expression, z-scores were calculated for each gene, and a Student's t-test compared expression levels in patients with high (top quartile) or low (bottom quartile) *ACVR1B* expression.

To identify breast cancer cell lines with low *ACVR1B* expression, *ACVR1B* mRNA expression was assessed in a panel of 51 breast cancer cell lines. Affymetrix U133+2 gene expression data (GSE12777)[88] were acquired from the Gene Expression Omnibus (GEO). Data were Microarray Suite 5 (MAS5) normalized and log2-transformed using the Affymetrix Gene Expression Console. Gene-specific expression probes were collapsed to the median using GenePattern[89], and z-scores were plotted (Supplementary Fig. 2a).

Gene expression profiles for breast and pancreatic cancer patients were retrieved from the GEO repository[90]. Expression values for genes of interest were log2-transformed and exported using the GEO2R analytical tool[91]. Kaplan–Meier survival curves for patient cohorts were generated using Kaplan–Meier Plotter (data last acquired on 1/19/2025; https://kmplot.com/analysis/)[92] or the International Cancer Genome Consortium (ICGC) PACA-AU dataset and GSE62452[93], with significance assessed by log-rank tests.

## Immunohistochemical staining
Formalin-fixed, paraffin-embedded 5 μm lung or pancreatic sections were deparaffinized and rehydrated. Endogenous peroxidase activity was inactivated by incubation with 2% $H_2O_2$. Antigen retrieval was performed using an antigen unmasking solution, followed by blocking with 8% BSA. Primary antibody was applied overnight at 4 °C. Sections were then incubated with a rabbit secondary antibody using a VEC-TASTAIN ABC-HRP Kit (Vector Laboratories). Immunoreactivity was detected using a 3,3′-diaminobenzidine (DAB) system (Vector Laboratories). The PHA-L staining protocol was developed and performed by the Duke Research Histology Lab.

## Luciferase assay

Cells were plated in six-well plates at $10^5$ cells/well. The following day, cells were transfected with 2.3 µg of pE2.1 luciferase reporter plasmid and 0.2 µg of pRL-SV40 Renilla plasmid. After 24 h, cells were treated with 10 µM SB431542 or 40 pM TGF-β1 for 20 h. Cells were lysed, and a dual-luciferase reporter assay (Promega) was performed according to the manufacturer's protocol.

## Binding and crosslinking assay

The binding and crosslinking assay was performed as described previously[80]. Briefly, cells were treated as specified and incubated with 40 pM [$^{125}$I] TGF-β (PerkinElmer). After incubation, the ligand was chemically cross-linked using 0.5 mg/ml disuccinimidyl suberate and quenched with 1 M glycine. The resulting complexes were separated by SDS–PAGE, and dried gels were analyzed using Typhoon phosphor-imaging and ImageJ software.

## Methionine pulse-chase labeling

The methionine pulse-labeling assay was performed as previously described[94]. Briefly, cells were incubated in pre-warmed pulse-labeling medium for 20 min. After incubation, cells were washed and transferred to pre-warmed chase medium for the indicated duration at 37 °C. Cells were collected, lysed, and immunoprecipitated with TβRII antibody. The immunoprecipitated samples were separated via SDS–PAGE, and the dried gels were analyzed using Typhoon phosphorimaging and ImageJ software.

## Flow cytometry

Cells were harvested using 0.05% trypsin and resuspended in FACS buffer (0.1% BSA in PBS). A total of $3.5 \times 10^5$ cells per sample were incubated in 1 ml FACS buffer with 2.5 µg/ml fluorescein-conjugated PHA-L (Vector Laboratories). Samples were rotated in a head-to-end manner at 4 °C for 30 min in the dark. After incubation, samples were pelleted by centrifugation, resuspended in FACS buffer, and analyzed. Three replicates were prepared for each condition and analyzed on a BD FACSCanto™ II (BD Biosciences). Data were processed using FlowJo™ v10 (BD Biosciences).

## Quantitative reverse transcription-PCR

RNA was isolated using Trizol™ Reagent (Thermo Fisher Scientific) and reverse-transcribed using an iScript cDNA Synthesis Kit (Bio-Rad). qRT-PCR was performed using iQSYBR Green Supermix and a Real-Time PCR Detection System (Bio-Rad). Cycle threshold values were determined using the Bio-Rad CFX Manager. Primers used in this study are listed in Supplementary Data 1.

## Quantitative proteomic analysis

The quantitative proteomic analysis was performed at The Center for Genomic & Computational Biology (GCB) at Duke University. PANC1-NTC and ALK4 KOs cells (4 replicates for each condition) were used. Cell lysates (~400 µl) were thawed, followed by the addition of 100 µl of 10% (w/v) SDS in 50 mM triethylammonium bicarbonate (TEAB) buffer. This was followed by probe sonication and then incubation at 80 °C to denature the proteins and facilitate complete lysis. Samples were then centrifuged, and the resulting pellet was undetectable. Protein (40 µg) was diluted with 5% (w/v) SDS in TEAB buffer, followed by the addition of 10 mM DTT and reduction at 80 °C for 10 min. After cooling, samples were alkylated with 25 mM iodoacetamide. SDS was removed, and samples were digested with Sequencing Grade Modified Trypsin (1:20, Promega) using an S-trap mini device (ProtiFi), according to the manufacturer's instructions. Recovered peptides were lyophilized to dryness and reconstituted in 40 µl of 1% trifluoroacetic acid (TFA) and 2% acetonitrile (MeCN).

One-dimensional liquid chromatography, tandem mass spectrometry (1DLC-MS/MS) was performed in a block-randomized order.

Samples were analyzed using a nanoACQUITY UPLC system (Waters) coupled to an Orbitrap Exploris 480 high-resolution, accurate-mass tandem mass spectrometer (Thermo Fisher Scientific) via a nanoelectrospray ionization source. Samples were first trapped on a Symmetry C18 180 µm × 20 mm trapping column (5 µl/min at 99.9/0.1 v/v H$_2$O/MeCN), followed by an analytical separation using a 1.7 µm ACQUITY HSS T3 C18 75 µm × 250 mm column (Waters) with a 90-min gradient of 5–30% MeCN with 0.1% formic acid at a flow rate of 400 nl/min and column temperature of 55 °C. Data collection was performed in a BoxCar data-independent acquisition (BoxCarDIA) mode with variable window BoxCar and DIA scans defined based on scouting runs. Each cycle utilized a full scan with a resolution of 120,000 resolution from $m/z$ 400 to 1200 with a normalized AGC target of 300% and 50 ms IT, two BoxCar scans at 120,000 resolution with 10 windows each, 100% AGC and auto IT, and a 25 vwin DIA scan with 30,000 resolution, AGC target of 1000%, a 60 ms IT and NCE of 30. Run-start internal calibration was enabled. Data were collected in centroid mode. For spectral library generation, a single data-dependent acquisition run was performed on an SPQC sample.

## Proteomic data analysis

A spectral library was generated using Spectronaut v. 14.10.201222.47784 (Biognosys). DIA files were converted to.htrms format using the HTRMS converter (Biognosys). Spectral library generation was performed using direct-DIA searching with Spectronaut Pulsar and a SwissProt database with *Homo sapiens* taxonomy, with the addition of BSA and porcine trypsin (20,385 entries). Database searching was conducted with default parameters, including trypsin/P specificity, up to two missed cleavages, N-terminal protein acetylation, Met oxidation, and Gln/Asn deamidation. This resulted in a study-specific library containing 101,6022 precursors, 79,042 modified peptides, and 5967 protein groups. Quantification was conducted using DIA-NN v.1.7.16 (https://github.com/vdemichev/DiaNN). Non-default analysis settings included: spectral library (exported from Spectronaut Pulsar search) and.fasta files (as described above); generation of the spectral library and MBR checked; neural network classification in double-pass mode; protein inference using protein names (from FASTA); and quantification strategy based on robust LC (high accuracy). Additional analyses were performed in R/RShiny using limma/edgR2 and Quickomics (https://github.com/interactivereport/Quickomics/).

## Functional annotation and pathway enrichment analysis

Pathway enrichment analysis was performed using the GSEA program (Broad Institute) following the developer's instructions. Expression of the top 125 most upregulated proteins was extracted for functional annotation and pathway enrichment analyses using DAVID Bioinformatics Resources 2021[95,96].

## Statistical analysis

Data from mouse experiments and patient samples were analyzed using the nonparametric two-tailed Mann–Whitney test. For multiple comparisons, nonparametric one-way ANOVA was performed using the Kruskal–Wallis test, followed by Dunn's multiple comparisons test. Survival data were analyzed using the log-rank test. In vitro experiments comparing two groups were analyzed using two-tailed Student's $t$-tests. For multiple comparisons within one independent variable, one-way ANOVA was followed by Tukey's test. For experiments involving two independent variables, two-way ANOVA with interaction testing was used. The correlation between the expression of two genes was assessed using nonparametric two-tailed Spearman's correlation. In all cases, data are presented as the mean ± standard error of the mean unless otherwise stated. Each experiment included at least three biological replicates. For Western blotting, one representative out of three independent replicates is shown in

the figures. All statistical analyses were performed using GraphPad Prism v9.0.0.

## Reporting summary

Further information on research design is available in the Nature Portfolio Reporting Summary linked to this article.

## Data availability

The proteomic data generated in this study have been uploaded to MassIVE with accession number MSV000091598 (https://massive.ucsd.edu/ProteoSAFe/dataset.jsp?task=699b8d9fa544472f8856ae14bf823ce6). Source data are provided with this paper. The remaining data are available within the Article, Supplementary Information or Source Data file. Source data are provided with this paper.

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

## Acknowledgements

This work was supported by the National Institutes of Health through grants NCI R21-CA198365 and R01-CA226925 (to G.C.B.), V2016-013 from the V Foundation for Cancer Research, and 133887-RSG-19-160-01-TBE from the American Cancer Society (to M.L.G.), and DCHS-20-PPC-014 from the New Jersey Commission for Cancer Research (to C.A.K.). We thank Duke Proteomics and Metabolomics Core Facility and Dr. Matthew Foster for conducting the proteomic studies and depositing the data. We thank Dr. Rebecca Bacon for performing the histological evaluation. We thank Duke Pathology Research Histology Lab and Dr. Zuowei Su for developing the IHC protocol for PHA-L stain.

## Author contributions

M.Z., J.C., and G.C.B. conceived and designed the research. G.C.B. supervised the project. M.Z., J.C., and G.C.B. designed the experiments. M.Z., J.C., M.C.L., J.G., X.S., and C.A.S. conducted the experiments. C.A.K., R.M., and M.L.G. performed the in silico data analysis. M.Z. J.C., M.C.L., J.G., C.A.S., C.A.K., R.M., M.L.G., E.T.O., and G.C.B. analyzed and interpreted the data. M.Z., J.C., and G.C.B. wrote the manuscript with input from coauthors.

## Competing interests

The authors declare no competing interests.
