## [Transparent Peer Review file · Nature Communications]

Loss of ALK4 Promotes Cancer Progression Through Regulating TGF- β Receptor N-glycosylation

Corresponding Author: Dr Gerard Blobe

Version 1:

Reviewer comments:

Reviewer #1

(Remarks to the Author)

In this interesting and thoroughly investigated manuscript, the authors identify ALK4 down regulation as a common feature of solid cancers and provide compelling evidence that this leads to enhancement of EMT/invasion/metastasis via increased TGF β signaling as a result of enhanced TGF RI/II surface retention from up-regulated Mgat5 N-glycosylation and its ligand galectin-3. As it has been previously shown that Mgat5 N-glycosylation/galectin-3 interactions regulate EMT/invasion/metastasis by decreasing TGF RI/II endocytosis (to increase TGF β signaling)¹⁻⁴, the novelty in this manuscript lies in the connection of ALK4 down regulation to Mgat5/galectin-3 up-regulation. However, as activin signaling is shown not to be impacted by ALK4 alteration, the mechanism by which this occurs is not clear. Additional investigations exploring this mechanism would greatly strengthen the manuscript. At the very least, the authors should look at non-canonical ALK signaling pathways⁵. This should include assessing short-term changes in canonical and non-canonical signaling pathways after acute blockage of ALK4. Other issues are below.

Other Major issues

- Fig. 8 and associated supplemental data needs to be done using MGAT5 KO cells (or inhibitors that specifically target branching like swainsonine or kifunensine), rather than NGL-1 and glucosamine. This should also be presented as the main figure and the NGL-1 and glucosamine data moved to supplemental. Unlike Mgat5 KO, NGL-1 and glucosamine inhibit N-glycan attachment by blocking LLO biosynthesis leading to significant ER stress and unfolded protein responses (UPR). This can complicate interpretation. In contrast, Mgat5 KO is a much subtler change to N-glycans, does not induce ER stress or UPR, and most importantly is the precise change identified by loss of ALK4.

- All metastasis models were via IV injection. A spontaneous metastasis model would strengthen the paper. At the very least the discussion and interpretation should be adjusted to reflect this.

- The authors demonstrate regulation of Mgat5 and its product, B1,6 branched glycans. However, no attempt is made to further characterize the glycosylation changes. The branching pathway enzymes are often regulated in a coordinated fashion, or metabolically. At the very least the authors should examine gene expression for enzymes in the branching pathway (eg Mgat1, 2, Mgat4a,b, mannosidase 1a,b,c, mannosidase II/IIx) and the hexosamine biosynthetic pathway. In addition glycomic analysis, or at least a panel of lectins in addition to L-PHA, should be used to determine if the changes are restricted to the B1,6 branch.

- the abstract, introduction and discussion should be updated to better reflect the published data that N-glycan branching and Mgat5 N-glycosylation regulate EMT/invasion/metastasis by decreasing TGF RI/II endocytosis (to increase TGF β signaling)¹⁻⁴. The concept of the galectin-glycoprotein lattice as an overall regulator of glycoprotein endocytosis should also be mentioned in the discussion⁶⁻⁸.

Other Minor Issues

- Supplemental figures should be in the order they are described in the text.
- P-values and statistical tests should be defined in figure legends.
- Fig. 2C should have statistical analysis and sample size.

- Sample size and number of times experiments repeated needs to be stated in the Figure legends.
- Quantitation of western blots should be provided throughout the manuscript to allow comparison between experiments.
- Blots should be consistently labelled, including addition of blot type to the panels (eg WB vs I-125-TGF- β cross-linking)
- Fig. 5H top panel, Fig S9B left panel and Fig S9D upper right panel should all have longer exposures.
- Unclear why switched to 293T cell in 5C,E ,F. Should be one of the cancer lines.
- Were any measures taken to ensure that observers were blinded to the experimental conditions, for example in the case of counting metastatic nodules or gross metastases in Fig2D-M. If so these should be described in the manuscript.
- Additional methodological detail should be provided. For example, in the TGF β treatment experiments, how was serum resting performed.
- What is the rationale for pre-treating for 72 hours with soluble activin receptor prior to 30min activin treatment? A shorter term incubation would be more appropriate.
- For Fig12H,I quantitation should be presented in bar graphs.
- Mgat5 mRNA levels should be measured in siMgat5 condition. The decrease in LPHA staining suggests poor inhibition of Mgat5. Or use Mgat5 KO as suggested above.
- Quantitation of Fig S14F is needed.

- 1 Dennis, J. W., Laferte, S., Waghorne, C., Breitman, M. L. & Kerbel, R. S. Beta 1-6 branching of Asn-linked oligosaccharides is directly associated with metastasis. *Science* 236, 582-585, doi:10.1126/science.2953071 (1987).
- 2 Granovsky, M. et al. Suppression of tumor growth and metastasis in Mgat5-deficient mice. *Nat Med* 6, 306-312, doi:10.1038/73163 (2000).
- 3 Partridge, E. A. et al. Regulation of cytokine receptors by Golgi N-glycan processing and endocytosis. *Science* 306, 120-124, doi:10.1126/science.1102109 (2004).
- 4 Lau, K. S. et al. Complex N-glycan number and degree of branching cooperate to regulate cell proliferation and differentiation. *Cell* 129, 123-134, doi:10.1016/j.cell.2007.01.049 (2007).
- 5 Zhang, Y. E. Non-Smad Signaling Pathways of the TGF- β Family. *Cold Spring Harb Perspect Biol* 9, doi:10.1101/cshperspect.a022129 (2017).
- 6 Demetriou, M., Granovsky, M., Quaggin, S. & Dennis, J. W. Negative regulation of T-cell activation and autoimmunity by Mgat5 N-glycosylation. *Nature* 409, 733-739, doi:10.1038/35055582 (2001).
- 7 Dennis, J. W., Nabi, I. R. & Demetriou, M. Metabolism, cell surface organization, and disease. *Cell* 139, 1229-1241, doi:10.1016/j.cell.2009.12.008 (2009).
- 8 Dennis, J. W., Lau, K. S., Demetriou, M. & Nabi, I. R. Adaptive regulation at the cell surface by N-glycosylation. *Traffic* 10, 1569-1578, doi:10.1111/j.1600-0854.2009.00981.x (2009).

Reviewer #2

(Remarks to the Author)

Reviewer #3

(Remarks to the Author)

The manuscript presents a curious observation that loss of the TGF- β type one receptor, Alk4, actually increases TGF- β 1 signaling and promotes a unique EMT event, capable of increasing invasion while maintaining proliferation and therefore drives overall metastasis. Mechanistically, the work goes on to show that loss of Alk4, increases TBRI and drives TBR1/TBR11 cell surface complexes. Proteomics is used nicely characterize the EMT phenotype that results following depletion of Alk4. This revealed a cohort of the glycoproteins. Use of N-linked glycosylation inhibitors prevented the increase in metastasis caused by loss of Alk4. Overall, the work presents extensive findings that support a highly novel concept that loss of Alk4, reduction of activin signaling and promotion of an epithelial phenotype is balanced by enhanced glycol signaling supporting a mesenchymal phenotype. Overall, these events result in the highly aggressive "hybrid" EMT. Overall, this is an extremely rigorous study that clearly presents highly innovative data that has the potential to lay the groundwork for a mechanistic understanding of EMT and metastasis not previously recognized. I would suggest a few minor critiques;

Several pieces of data are presented establishing that Alk4 is decreased in breast and PANC and that this is associated with disease progression. Some discussion could be offered as to what events lead to the loss of Alk4, i.e. genetic, transcriptional, post-transcriptional.

NGI-1 is a highly insoluble compound. Some level of PD from the in vivo studies (i.e. changes in TBR-I or other protein glycosylation) would help support on-target effects of the observed anti-tumor activity.

The use of the term metastasis in reference to pulmonary tumors established by tail vein inoculation should be modified to pulmonary lesions or something similar.

Reviewer #4

(Remarks to the Author)

In this manuscript the authors attempted to demonstrate that the loss of Activin receptor-like kinase 4 (ALK4) plays an instrumental role in breast and pancreatic cancer progression and metastasis. Through a comprehensive analysis of several public datasets, they showed that ACVR1B (the gene encoding ALK4) was significantly mutated or lost in breast and pancreatic cancers, and that low expression levels of ACVR1B were associated with the worst prognosis in these patients. Subsequent functional experiments showed that knocking down ALK4 expression in breast or pancreatic cell lines promoted anchorage-independent growth, cell migration/invasion, epithelial-mesenchymal transition, and metastasis. Mechanistically, loss of ALK4 was associated with increased TGF- β /Smad canonical signaling, presumably owing to increased N-glycosylation and stability of both T β RI and T β RII at the cell surface. Gene expression analysis revealed that loss of Alk4 culminated in increased expression of MGAT5 and galectin-3, which together facilitate the N-glycosylation of T β RI and T β RII and thereby activate TGF- β signaling. While at a first glance the hypothesis that the loss of ALK4 contributes to cancer progression might seem appealing, the study does not provide compelling evidence to support such claim. It also lacks the conceptual advance needed for establishing the relevance of this mechanism to the human diseases. Indeed, the vast majority of experiments are superficial in nature, relying mostly on in vitro cell systems, which could lead to all sorts of artifacts. Therefore, one would anticipate that this study would deliver an incremental impact in the field of breast and pancreatic cancers. A few issues are listed below.

The authors relied heavily on several public datasets to show that Alk4 is lost in breast and pancreatic cancers, with low levels of Alk4 correlating with poor prognosis. The authors might want to consider corroborating these data using qRT-PCR or immunohistochemistry with tissue microarrays. In any case, considering the possibility that Alk4 expression is lost in breast and pancreatic cancers, one would wonder what mechanisms could contribute to this phenomenon in the cancerous samples that do not display any mutation or loss of copy numbers? Is there any other genetic or epigenetic event that could compromise ALK4 expression? This information is important for understanding the exact contribution of ALK4 to cancer progression and metastasis.

The expression levels of ALK4 seem to be inversely correlated with the aggressiveness and metastatic potential of breast and pancreatic cancer cell lines. However, this correlation is based on very artificial invasion assays in vitro. To support their claim, the authors might want to consider characterizing these cell lines in vivo, both in terms of tumor growth and metastatic potential. Based on the literature, it is perplexing to see that MIA Paca-2 and AsPC-1, which display poorly differentiated architectures, are less aggressive than Panc1, which display a well differentiated epithelial architecture. In any event, to provide compelling evidence that the loss of ALK4 contributes to cancer progression, it is imperative to conduct in vivo experiments with autochthonous mouse models of breast and pancreatic cancer. It would be straightforward to delete AcvR1B in mouse models of breast and pancreatic cancer (such as in MMTV-PyMT or KC mice with conditional expression of KrasG12D in the pancreatic epithelium). These experimental approaches must be presented at his level of publication.

ALK4 seems to induce expression of T β RI and T β RII at the surface by a mechanism dependent on its kinase activity. What is the substrate of ALK4 under these conditions? This information is important for understanding how ALK4 fulfills its anti-tumor suppressive function in breast and pancreatic cancers. Also important is what signaling pathway functions downstream of ALK4 to suppress cancer progression and metastasis? Without providing this kind of data, the impact of the study will be just incremental.

Loss of ALK4 correlates with increased expression of MGAT5 and galectin-3, which could provide a mechanistic explanation as to how ALK4 regulates the N-glycosylation and thereby stability of T β RI and T β RII at the cell surface. There are at least three issues with this observation. First, the mechanism leading to the accumulation MGAT5 and galectin-3 in cells deficient for Alk4 was not explored. Second, compelling evidence that T β RI and T β RII are direct substrate of MGAT5 and galectin-3 was not presented. Third, compelling evidence that MGAT5 and galectin-3 functions downstream of ALK4 in vivo under physiological conditions is missing. The latter could be easily achieved through using mice or cells co-deleted of MGAT5 /galectin-3 and Alk4 and treated with the two inhibitors NGI-1 and Glucosamine.

Overall, the study lacks the mechanistic depth for firmly establishing a role for ALK4 in suppressing cancer progression and metastasis. It also lacks the mechanistic depth for firmly establishing how ALK4 regulates the stability of T β RI and T β RII at the cell surface and attendant activation of TGF- β /Smad signaling pathway. Expression of ALK4 at the cell surface might simply rely on the same mechanism as does localization of T β RI and T β RII, so one conceivable mechanism would be that ALK4 might compete with T β RI and T β RII for the machinery that drives their cell surface localization.

Reviewer #5

(Remarks to the Author)

In this manuscript, Zhang et al present three major findings: 1) low ALK4 (the type I receptor for activin) expression correlates with poor prognosis in human breast and pancreatic cancer patients as well as EMT, invasion and metastasis of cancer cells; 2) knock-down of ALK4 promotes TGF- β signaling, its cell surface receptor levels, and receptor glycosylation; and 3) the effects of ALK4 knock-down are likely mediated through increase in galectin-3 and MGAT5 (a N-acetylglucosaminyltransferase) function in promoting glycosylation of TGF- β receptors. Although each claim was supported individually by a large body of data, the mechanism that links loss of ALK4 function to Gal-3 and Mgat5 transcription and thereby glycosylation of TGF- β receptors is missing. This deficiency in the authors' thesis is particularly troublesome in light of their own revelation that blocking activin signaling via a soluble Fc fragment of ActRIIB showed no effect on the

enhancement of TGF- β signaling by ALK4 knockdown. Another major problem with this manuscript is with their data/experiment description, which are insufficient, imprecise/inaccurate, and often hard to follow. While this second problem can certainly be readily rectified through a careful rewrite, the lack of the underlying mechanism or a reasonable conceptual connection makes this work incomplete for publication. Some specific issues are as follows.

1. Comparison of ALK4 levels in various cancer cell lines is not meaningful or superficial at best (sFig.2C-D), as underlying causal factors vary from one cell type to another. For the same reason, comparison in sFig.9A is also problematic, e.g. ALK4 level is similar between 168FARN and 4T07 (sFig 2D) but cell surface TGF β R are quite different (sFig 9A). Moreover, the data were too scarce to support the correlation.
2. Authors made several lines of stable cells expressing shALK4 or ALK4 gRNA. These lines were used interchangeably throughout the manuscript. It is not clear why some experiments using the shALK4 lines while others using the ALK4KO lines. Sometimes panels were labelled with ALK4KO while legend said otherwise.
3. Experiments in (Fig.2G-M) were inadequately described. Legend 2G says the cells were implanted in “the tail of pancreas”. Were they implanted in the pancreas or injected in tail vein? It appears that authors attempt to show that silencing ALK4 decreased primary tumor growth in the pancreas but increased invasiveness and distant metastasis, which is in line with the bimodal roles of TGF- β in cancer.
4. It is known that both activin and TGF- β are capable of inducing Smad2 and Smad3 phosphorylation. And it appears that ALK4 KO by itself consistently induced Smad2 phosphorylation, and enhanced TGF- β -mediated Smad2/3 phosphorylation (Fig 4A-C). Again, the legend in Fig 4C was different from the panel label. These experiments were done in cancer cells, phosphorylation of Smad2/3 should also be examined in cancer models.
5. “Clinically” is inappropriately used in describing results under “loss of ALK4 induces EMT...”, as the finding described was from analysis of human samples, not clinical studies. Fig.4E is poorly described and not clear what the conclusion was reached, and it is certainly not a dot-plot.
6. Fig. 4H, why DN-TbRII did not block TGF- β induction of Smad2 phosphorylation (shNTC data group)?
7. The sActRIIB-Fc experiments appear to show that effects of ALK4 KO on TGF- β signaling not due to activin signaling per se. Since ALK4 was once thought to mediate TGF- β signaling and it does possess some effect, could it be that there is a negative feedback loop connecting ALK4 to TGF- β signaling?
8. Fig.5A lacks controls showing ALK4 levels after KO. Fig.5D missing ALK4 expression control. sFig.9E is a haphazard control for Fig.5D.
9. Galectin-3 was singled out based on mapped interactions with TGF- β targeted genes, coreceptors, FN1, MMP2, CD44, and endoglin in the STRING analysis. However, there are quite a few other nodes in this interaction network that have not been examined or ruled out. No compelling reason was given to proceed with Gal-3. It appears to be cherry picked.
10. More importantly, authors showed that knockdown or KO ALK4 increased Gal-3 and Mgat5 transcription, the increased Gal-3 and Mgat5 promoted glycosylation of TGF- β receptors and increased TGF- β signaling, however, siTbRI had no effect on Gal-3 and Mgat5 transcription (sFig 12). Then how did shALK4 (or ALK4KO) enhance Gal-3 and Mgat5 transcription? Did Smad2 or Smad3 play a role in their transcription?

Version 2:

Reviewer comments:

Reviewer #1

(Remarks to the Author)

The authors have made a solid attempt to address our concerns, but some minor issues need to be addressed before publication:

Concern 1: Recommend to put ETS1 data in main figure, and cite known regulation of Mgat5 by ETS1

Concern 2: the new Mgat5KO data is convincing, but recommend that TGF β signaling be assessed as in Fig8A. Also again strongly recommend that Fig. 8 should be moved to supplemental figures and the new Mgat5KO data be its own Figure.

Concern 3: The spontaneous metastatic data is limited (see issues below) and at the very least the interpretation should acknowledge that the effects on metastasis were less clear.

In Fig2k - Statistical analysis for this panel is confusing. 3/8 is 37.5% not 45%. The figure legend states that this was done by Mann-Whitney. Statistical analysis should be re-done as a contingency table and clarifying whether the same mouse had multiple mets or all mice with mets were different mice. Alternatively, the number of mets per mouse could be shown leaving aside location.

In FigS3g, lung mets should be quantitated.

In FigS4, ascites can reflect local invasion rather than metastasis and the two should not be grouped together.

sFIG4a: x-axis should say days not months

Concern 4: Please clarify the lectin array methodology in the manuscript. Is this using lysate or cells?

Figure FigL1 should be included in the paper with a clear definition of the mass spec assay used. Is this UDP-HexNAc or UDP-GlcNAc? Discuss discrepancy between UDP-GlcNAc and lectin array. Why so few changes in lectin binding? Also, why no increases in Galectin staining as expected?

Concern 5: The authors have largely updated the abstract, introduction and discussion to better reflect the published data that N-glycan branching, but a sentence in the abstract should be changed (as highlighted in red) below to be more accurate:

“Consistent with prior observation that galectin-3 preferentially binds to MGAT5-modified glycoproteins to stabilize cell surface receptors like TGF- β receptors, we found that ALK4 loss enhanced MGAT5-mediated glycosylation of TGF- β receptors,…”

Concern 6-8: Addressed.

Concern 9: Some figure legends state that in vitro experiments were done with 3 biological replicates, but do not state if the experiments were independently replicated. This should be clarified. Some figure legends state the number of independent experiments but others do not. This should be clearly stated in all figure legends.

Concern 10: Some WBs have been quantitated in the figures, others have been presented as a supplemental excel file. It would be best if all WB quantitation was present within the figures (at least the main figures).

Concern 13: Figure L2 should be in the manuscript in the main figure as Fig5d.

Concern 14: We could not find blinding in the methods section, this should be corrected. In addition, the figure legends should state when blinding was performed.

Other Minor concerns:

Figure 1 legend is incomplete or cut off in the PDF.

The figures often refer only to “KO”. These should be fixed to identify the specific KO (eg “ALK4 KO”)

Reviewer #3

(Remarks to the Author)

The authors have addressed my previous, minor, critiques. The authors should be commended on completion of this highly impactful, rigorous, and scientifically interesting manuscript.

Abstract line 44; consistent with prior observations

Reviewer #4

(Remarks to the Author)

This second revision focuses solely on the original concerns raised by this reviewer.

The observation that ALK4 may function as a tumor suppressor in breast and pancreatic cancers by antagonizing T β RI/T β RII is intriguing and warrants further mechanistic and translational studies. However, the level of innovation may be limited, as T β RI and T β RII themselves can act as tumor suppressors depending on the physiopathological context.

Additional unresolved issues include the absence of robust mechanistic studies, which may stem from the use of insufficiently advanced technologies. Overall, this paper does not present the conceptual advancements necessary to drive new discoveries suitable for publication in a high-profile journal.

Reviewer concern 1: While at a first glance the hypothesis that the loss of ALK4 contributes to cancer progression might seem appealing, the study does not provide compelling evidence to support such claim. It also lacks the conceptual advance needed for establishing the relevance of this mechanism to the human diseases. Indeed, the vast majority of experiments are superficial in nature, relying mostly on in vitro cell systems, which could lead to all sorts of artifacts.

Author response: In this manuscript, we used different knockdown and knockout models including transient knockdown using siRNA, stable knockdown using shRNA, CRISPR knockout and Cre recombinase mediated genetic deletion to observe consistent effects of ALK4 loss on cancer progression. We further used unbiased proteomic analysis to establish mechanistic leads, and validated this mechanism in epistatic experiments, including in vivo studies. In addition to extensive in vitro analysis, we performed in vivo mouse studies including tail vein inoculation and orthotopically injection using 4 different cell lines with both loss and gain of function models and now add a GEMM. We realize that any scientific method has its limitations, but utilizing multiple complementary approaches, we have provided consistent results.

New Reviewer critique: The authors have made significant efforts to address the concerns raised by both this reviewer and others. However, compelling evidence supporting ALK4's role as a tumor suppressor in breast and pancreatic cancers remains insufficient. Numerous advanced technologies could be leveraged to elevate this potentially intriguing study, providing the depth needed to justify its publication in a high-profile journal.

Reviewer concern 2: The authors relied heavily on several public datasets to show that ALK4 is lost in breast and pancreatic cancers, with low levels of ALK4 correlating with poor prognosis. The authors might want to consider corroborating these data using qRT-PCR or immunohistochemistry with tissue microarrays. In any case, considering the possibility that ALK4 expression is lost in breast and pancreatic cancers, one would wonder what mechanisms could contribute to this phenomenon in the cancerous samples that do not display any mutation or loss of copy numbers? Is there any other genetic or epigenetic event that could compromise ALK4 expression? This information is important for understanding the exact contribution of ALK4 to cancer progression and metastasis.

Author response: We agree with the reviewer's concern. For the presented in silico analysis, we investigated in total 192 PAAD GEO datasets and 1,464 BRCA GEO datasets, the presented datasets meet the following criteria: 1) datasets that contain gene expression profile from normal tissue; and 2) datasets that contain a minimum of 3 samples in each group (normal tissue and tumor tissue). To corroborate our in silico data, as the reviewer suggested, we acquired RNA samples from patients with pancreatic primary tumors and adjacent normal tissue from Duke Biorepository & Precision Pathology Center. Consistent with published database, ACVR1B expression is significantly lower in tumor tissues (new Figure 1c). While somatic mutation of ALK4 have been identified in a small percentage of breast and pancreatic cancers, there are other mechanisms contributing to loss of ALK4 function, including loss of copy number/loss of heterozygosity. In addition, we now provide evidence supporting epigenetic silencing of the ALK4 gene, including data demonstrating that treating pancreatic and breast cancer cells with the DNA methylation inhibitor 5-azacytidine (5-AZA) increased ALK4 expression, and that methylation of the ALK4 promoter in tumor samples correlates with decreased ALK4 expression (new sFigure 2c-h). These data and an overall discussion of mechanisms of loss of ALK4 function have been added to the revised discussion.

New Reviewer critique: The authors have addressed this concern by incorporating new data from patient samples to validate the analysis of publicly available datasets.

Reviewer concern 3: The expression levels of ALK4 seem to be inversely correlated with the aggressiveness and metastatic potential of breast and pancreatic cancer cell lines. However, this correlation is based on very artificial invasion assays in vitro. To support their claim, the authors might want to consider characterizing these cell lines in vivo, both in terms of tumor growth and metastatic potential.

Author response: We appreciate this concern of the reviewer. To evaluate the metastatic potential of these cell lines, we performed tail vein inoculation using PANC-1, HPNE, Mia Paca 2 and ASPC1 cells expressing luciferase. Briefly, half million cells of each line were injected into 8 NSG mice and the average bioluminescence was detected every week for up to 6 weeks. As shown in Figure L3, we observed that ASPC1 was the most aggressive cells line, followed by Mia Paca 2 and then HPNE/PANC-1. Accordingly, we agree with the reviewer's concern and have deleted this correlative data from the revised manuscript.

New Reviewer critique: The authors have not addressed this concern, which is crucial for demonstrating that ALK4 loss is associated with metastasis. Given that they have already conducted in vivo experiments using cell lines with varying ALK4 expression levels, it should be relatively straightforward to evaluate their metastatic potential by analyzing organs from the mice that received these cells (already done). Failing to do so could undermine the validity of their working model.

Reviewer concern 4: To provide compelling evidence that the loss of ALK4 contributes to cancer progression, it is imperative to conduct in vivo experiments with autochthonous mouse models of breast and pancreatic cancer. It would be straightforward to delete AcvR1B in mouse models of breast and pancreatic cancer (such as in MMTV-PyMT or KC mice with conditional expression of KrasG12D in the pancreatic epithelium). These experimental approaches must be presented at his level of publication.

Author response: We agree with this concern, and in response, we have incorporated in vivo experiments with autochthonous mouse models of pancreatic cancer, with oncogenic KRASG12D driven by Pdx1-Cre, with or without ALK4 (ALK4^{fl/fl} loss), which demonstrated decreased survival and increased spontaneous metastases with loss of ALK4 (new sFigure 4). The corresponding GEMM mouse data is now described and discussed in the revised manuscript.

New Reviewer critique: The authors have not addressed whether ALK4 ablation in the MMTV-PYMT mouse model (or an equivalent model) suppresses mammary tumor progression and metastasis. Given that they already have floxed ALK4 mice, this experiment should be relatively straightforward to conduct. However, the authors conducted experiments using the KC mouse model of pancreatic cancer and demonstrated that Alk4 deletion in the KrasG12D background accelerates pancreatic cancer progression and metastasis. They should quantify the

H&E staining for all examined pancreatic tissues and provide at least a histological analysis of the organs affected by metastasis (e.g., liver, lung, etc.). Additionally, to further validate their working model, they should present evidence of increased TGF β signaling, elevated expression of MGAT5, and upregulated T β RI and T β RII expression, along with enhanced N-glycan modifications of these receptors. This is the most robust model system presented in the study, and they should fully leverage its potential.

Reviewer concern 5: ALK4 seems to induce expression of T β RI and T β RII at the surface by a mechanism dependent on its kinase activity. What is the substrate of ALK4 under these conditions? This information is important for understanding how ALK4 fulfills its anti-tumor suppressive function in breast and pancreatic cancers. Also important is what signaling pathway functions downstream of ALK4 to suppress cancer progression and metastasis? Without providing this kind of data, the impact of the study will be just incremental.

Author response: We appreciate this concern. While we have not been able to define the precise substrate directly phosphorylated by ALK4 to mediate these effects, we have investigated the mechanism by which loss of ALK4 mediates its effects on increasing MGAT5 and galectin-3. As another reviewer suggested, we first have investigated whether non-canonical ALK signaling was affected after acute blockage of ALK4 via siRNA mediated silencing of ALK4 expression. As shown in new sFigure 16, siRNA mediated silencing of ALK4 expression increased p38 phosphorylation, without altering ERK or Akt signaling (sFigure 16a-c). However, inhibiting the p38 pathway using the small molecule inhibitor SB203580 did not affect TGF- β signaling or expression of MGAT5/galectin-3 (sFigure 16d-f). We also investigated whether ALK4 loss mediated increases in TGF- β signaling were responsible for upregulating Mgat5/galectin-3 to induce expression of T β RI and T β RII and demonstrated that they were not (sFigure 15). Finally, we investigated whether ALK4 loss upregulates Mgat5/galectin-3 through a specific transcription factor. As shown in new sFigure 17, we found that ETS1 expression is significantly increased by ALK4 loss, and that silencing of ETS1 decreased MGAT5 induction mediated by ALK4 loss expression, supporting a role of ETS1 downstream of ALK4 loss.

New Reviewer critique: The response to this issue is insubstantial and fails to provide convincing evidence that ETS1 is indeed the transcription factor regulating MGAT5 expression. While existing literature supports the role of ETS1 in regulating MGAT5 expression, the authors have not investigated how ALK4 loss leads to increased ETS1 expression. Consequently, the approach taken to address this concern remains superficial. The authors could strengthen their mechanistic insights by leveraging their KC and AKC mouse models to perform omics analyses (e.g., RNA-Seq, ATAC-Seq, spatial transcriptomics) to establish a more robust mechanistic framework, particularly regarding the interactions between ALK4, MGAT5, T β RI, T β RII, and ETS1.

Reviewer concern 6: Loss of ALK4 correlates with increased expression of MGAT5 and galectin-3, which could provide a mechanistic explanation as to how ALK4 regulates the N-glycosylation and thereby stability of T β RI and T β RII at the cell surface. There are at least three issues with this observation. First, the mechanism leading to the accumulation MGAT5 and galectin-3 in cells deficient for Alk4 was not explored. Second, compelling evidence that T β RI and T β RII are direct substrate of MGAT5 and galectin-3 was not presented. Third, compelling evidence that MGAT5 and galectin-3 functions downstream of ALK4 in vivo under physiological conditions is missing. The latter could be easily achieved through using mice or cells co-deleted of MGAT5/galectin-3 and Alk4 and treated with the two inhibitors NGI-1 and Glucosamine.

Author response: We have investigated the mechanism by which loss of ALK4 mediates its effects on increasing MGAT5 and galectin-3, as addressed in Concern 5. In terms of whether T β RI and T β RII are direct substrates of MGAT5 and galectin-3, this connection was established by Partridge et al [8], so we did not re-address here. We did establish that MGAT5 modified branched N-glycan were increased on T β RII (Figure 7d) and that internalization of T β RII level is decreased in the absence of ALK4 (Figure 7g) to further support this connection. To address whether MGAT5 is downstream of ALK4, we developed isogenic MGAT5 CRISPR KO cells using the PANC-1 NTC and ALK4 KO cells respectively. The knockout of MAGT5 was confirmed using flow cytometry (new sFigure 14c). As shown in new Figure 7h-k, we observed a significant decrease in anchorage-independent growth and development of pulmonary lesions when MGAT5 is depleted in ALK4 KO cancer models, supporting a role for MGAT5 mediated branched N-glycan modifications in mediating the effects downstream of ALK4 loss.

New Reviewer critique: This concern was addressed only partially and indirectly. To rigorously resolve this issue, the authors should reconstitute MGAT5-deleted cells with a catalytically inactive MGAT5 mutant that is incapable of mediating branched N-glycan modifications on T β RI/T β RII and demonstrate that this mutant fails to rescue the anchorage-independent growth and metastatic phenotypes. Without this approach, it would be challenging to definitively establish that MGAT5 functions through its ability to mediate branched N-glycan modifications.

Reviewer concern 7: Overall, the study lacks the mechanistic depth for firmly establishing a role for ALK4 in suppressing cancer progression and metastasis. It also lacks the mechanistic depth for firmly establishing how ALK4 regulates the stability of T β RI and T β RII at the cell surface and attendant activation of TGF- β /Smad signaling pathway. Expression of ALK4 at the cell surface might simply rely on the same mechanism as does localization of T β RI and T β RII, so one conceivable mechanism would be that ALK4 might compete with T β RI and T β RII for the machinery that drives their cell

surface localization.

Author response: We believe we have presented strong evidence for ALK4 in suppressing cancer progression, and mechanistic insight into how ALK4 regulates the stability of T β RI and T β RII, via MGAT5 and galectin-3. In terms of whether ALK4 might compete with T β RI and T β RII for the machinery that drives their cell surface localization, given the similarity between ALK4 and T β RI we considered that possibility, however our proteomic data and preliminary data support the upregulation of a broad spectrum of glycosylated cell surface proteins/receptors, including integrins in PANC-1 cells and VEGFR2 in endothelial cells (Figure 6g and Figure L4), making this less likely. This is now discussed.

New Reviewer critique: These observations are not particularly relevant to the proposed working model. With that being said, the mechanistic studies necessary to provide definitive evidence that ALK4 suppresses TGF β signaling and thereby prevents the progression of pancreatic and breast cancers are either unconvincing or entirely lacking. Given that previous studies have identified ALK4 as a receptor for TGF β , conducting thorough mechanistic investigations is crucial for the validity of this study. Additionally, a major issue remains: the inactivation of T β RII has been shown to accelerate pancreatic cancer in the KC mouse model of PDAC. How do the authors reconcile their findings with

Reviewer #5

(Remarks to the Author)

The authors made substantial improvement in the revised manuscript, and addressed most issues raised previously. With regard to the missing mechanism that accounts for the increased Mgat5 transcription/TbR glycosylation due to loss of ALK4, the authors performed some additional experiments to examine noncanonical TGF-b signaling. However, the results were inconclusive and the newly reported link to ETS1 is insufficiently informative. Given that a role of ActRIIB was also ruled out, the authors should acknowledge as such at least in the discussion to leave this issue open for future exploration.

Version 3:

Reviewer comments:

Reviewer #1

(Remarks to the Author)

All major issues have been addressed. Two minor presentation/labelling issues are still recommended to be fixed prior to publication

Concern 10 - again strongly recommend that all blots (other than shift in bands) be presented with quantitation.

Fig 7h,i,j still need to have the 'KO' label specified as 'ALK4 KO'.

Reviewer #6

(Remarks to the Author)

The authors have sufficiently addressed the comments of Reviewer 4. Because the evidence for a tumor suppressive role for ALK4 in breast cancer is much weaker than in PDAC, the authors should emphasize the limitations of current evidence in support of a tumor suppressive role in that tumor type, despite positive results in breast cancer cell lines.

Response to referees

Point-by-point response

Reviewer #1

We thank the reviewer for finding that the manuscript is “interesting and thoroughly investigated” and provides “compelling evidence” supporting its central claims, including that the “novelty in this manuscript” lies in the connection of ALK4 down regulation to Mgat5/galectin-3 up-regulation.

Concern 1: Additional investigation exploring the mechanism by which ALK4 loss upregulates Mgat5/galectin-3. would greatly strengthen the manuscript. At the very least, the authors should look at non-canonical ALK signaling pathways. This should include assessing short-term changes in canonical and non-canonical signaling pathways after acute blockage of ALK4.

We appreciate this concern, and as suggested, we investigated whether non-canonical ALK signaling was affected after acute blockage of ALK4 via siRNA mediated silencing of ALK4 expression. As shown in new sFigure 16, siRNA mediated silencing of ALK4 expression increased p38 phosphorylation, without altering ERK or Akt signaling (sFigure 16A-C). However, inhibiting the p38-MAPK pathway using the small molecule inhibitor SB203580 did not affect TGF- β signaling (sFigure 16D) or expression of MGAT5/galectin-3 (sFigure 16E, F). We also investigated whether ALK4 loss mediated increases in TGF- β signaling were responsible for upregulating Mgat5/galectin-3, and demonstrated that they were not (sFigure 15). Finally, we investigated whether ALK4 loss upregulates MGAT5/galectin-3 through a specific transcription factor. As shown in new sFigure 17, we found that expression of the transcription factor ETS1 is significantly increased by ALK4 loss, and that silencing of ETS1 decreased MGAT5 induction mediated by ALK4 loss expression, supporting a role of ETS1 downstream of ALK4 loss.

Concern 2: Fig. 8 and associated supplemental data needs to be done using MGAT5 KO cells (or inhibitors that specifically target branching like swainsonine or kifunensine), rather than NGI-1 and glucosamine. This should also be presented as the main figure and the NGI-1 and glucosamine data moved to supplemental. Unlike Mgat5 KO, NGI-1 and glucosamine inhibit N-glycan attachment by blocking LLO biosynthesis leading to significant ER stress and unfolded protein responses (UPR). This can complicate interpretation. In contrast, Mgat5 KO is a much subtler change to N-glycans, does not induce ER stress or UPR, and most importantly is the precise change identified by loss of ALK4.

We appreciate this concern, and as suggested, we developed two MGAT5 CRISPR KO lines using the PANC-1 NTC and ALK4 KO cells respectively. The knockout of MGAT5 was confirmed using flow cytometry (new sFigure 14c). As the reviewer recommended, using in vitro and in vivo analysis, we investigated whether loss of ALK4 promotes cancer progression through branched N-glycan catalyzed by MGAT5. As shown in new Figure 7h-k, we observed a significant decrease in anchorage-independent growth and development of pulmonary lesions when MGAT5 is depleted in ALK4 KO cancer models, supporting a role for MGAT5 mediated branched N-glycan modifications in mediating the effects of ALK4 loss.

Concern 3: All metastasis models were via IV injection. A spontaneous metastasis model would strengthen the paper. At the very least the discussion and interpretation should be adjusted to reflect this.

We agree that spontaneous metastasis models more accurately reflect the entire metastatic process. We had 2 such models in the original manuscript, an orthotopic pancreatic cancer model using HPNE shNTC and shALK4 cells (Figure 2 g-k) and an orthotopic breast cancer model using LM2 pBabe and LM2 pBabe ALK4 cells (sFigure 3 c-i). Both studies support the role of ALK4 in suppressing tumor metastasis. We have increased the clarity in describing these experiments in the revised manuscript. In addition, for all tail vein injection models, we now refer to the resulting pulmonary nodules as pulmonary lesions.

To complement these studies, we also included a GEMM of spontaneous pancreatic cancer and pancreatic cancer metastasis, with oncogenic KRAS^{G12D} driven by Pdx1-Cre, with or without ALK4 (ALK4^{fl/fl} loss), which demonstrated decreased survival, increased incidence of adenocarcinoma and increased development of ascites with loss of ALK4 (new sFigure 4).

Concern 4: The authors demonstrate regulation of Mgat5 and its product, B1,6 branched glycans. However, no attempt is made to further characterize the glycosylation changes. The branching pathway enzymes are often regulated in a coordinated fashion, or metabolically. At the very least the authors should examine gene expression for enzymes in the branching pathway (eg Mgat1, 2, Mgat4a,b, mannosidase 1a,b,c, mannosidase II/IIx) and the hexosamine biosynthetic pathway. In addition, glycomic analysis, or at least a panel of lectins in addition to L-PHA, should be used to determine if the changes are restricted to the B1,6 branch.

We appreciate this concern, and as suggested, we have investigated whether loss of ALK4 affects expression of other enzymes involved in modifying N-glycan, and whether these effects are specific to the B1,6 branch. We found that ALK4 loss specifically upregulates

expression of MGAT5 in both PANC-1 and MDA-MB-231 cells as shown in new sFigure 13d-e. Consistent with the expression data, using a lectin array, we observed that among all lectins that recognized N-glycans, only the intensity of lectin PHA-L was significantly increased ($FC > 2$, $p \text{ value} < 0.05$) by loss of ALK4 (new sFigure 13f-g).

In addition, we found that loss of ALK4 specifically increased expression of GFPT2 at both the protein and RNA level in PANC-1 and MDA-MB-231 cells (Figure L1 A-C). GFPT1/2 are important rate-limiting enzymes in the hexosamine biosynthetic pathway (HBP) [1] which produces uridine diphosphate-N-acetyl glucosamine, UDP-GlcNAc, a key metabolite that is used for N/O-linked glycosylation. Indeed, using mass spectrometry, we found that the level of UDP-GlcNAc is significantly increased in both PANC-1 and MDA-MB-231 cells with ALK4 loss (Figure L2D). This is consistent with the previous finding that cell surface receptors with four or less NXS/T sites require higher UDP-GlcNAc and branching to be retained by the galectin lattice [2].

The reason we specifically focused on MGAT5 in this manuscript is because galectin-3, as a potential lead acquired from our proteomic analysis, has the highest binding affinity with MGAT5 catalyzed mediation of β 1-6-GlcNAc branching [3, 4]. In this manuscript, we primarily focused on the function of ALK4, specifically its novel role in regulation TGF- β receptor glycosylation and cell surface stability. Therefore, we think that reporting ALK4 regulation more broadly is beyond the scope of the current studies. We are currently investigating ALK4 regulation of the HBP and metabolism in ongoing work.

Figure L1: Loss of ALK4 increases production of UDP-GlcNAc level in breast and pancreatic cancer. (A-B) qRT-PCR analysis of mRNA expression for key enzymes of hexosamine biosynthetic pathway in PANC-1 (A) and MDA-MB-231 (B) NTC and ALK4 KO cells, with GAPDH as the reference control. (C) Western blot analysis of GFAT2 and β -actin levels in PANC-1 and MDA-MB-231 NTC and ALK4 KO cells. (D) Bar graph showing total amount of UDP-GlcNAc in PANC-1 NTC and ALK4 KO cells and MDA-MB-231 NTC and ALK4 KO cells. For comparison between two groups, data were analyzed using two-tail Student's t tests. * $p < 0.05$, **** $p < 0.0001$.

Concern 5: The abstract, introduction and discussion should be updated to better reflect the published data that N-glycan branching and Mgat5 N-glycosylation regulate EMT/invasion/metastasis by decreasing TGF β RI/II endocytosis (to increase TGF β signaling)1-4. The concept of the galectin-glycoprotein lattice as an overall regulator of glycoprotein endocytosis should also be mentioned in the discussion6-8.

These previous findings have been further described in the abstract, introduction and discussion, and the corresponding references have been cited.

Other Minor Issues

Concern 6: Supplemental figures should be in the order they are described in the text.

This has been addressed throughout the manuscript.

Concern 7: P-values and statistical tests should be defined in figure legends.

P value and the statistical tests used have been defined in the figure legends as requested.

Concern 8: Fig. 2C should have statistical analysis and sample size.

The p value and sample size for Fig. 2C has been added.

Concern 9: Sample size and number of times experiments repeated needs to be stated in the Figure legends.

Sample size and the number of times experiments repeated are stated in the figure legends.

Concern 10: Quantitation of western blots should be provided throughout the manuscript to allow comparison between experiments.

We have performed quantification all western blots throughout the manuscript and this is now included in Supplemental Table 4, or in the Figures and Supplemental Figures.

Concern 11: Blots should be consistently labelled, including addition of blot type to the panels (eg WB vs I-125-TGF-b cross-linking)

All blots have been labelled.

Concern 12: Fig. 5H top panel, Fig S9B left panel and Fig S9D upper right panel should all have longer exposures.

We have adjusted the exposure of all panels as requested.

Concern 13: Unclear why switched to 293T cell in 5C,E ,F. Should be one of the cancer lines.

Our goal for performing these experiments was to determine whether ALK4 kinase activity is involved in regulating receptor stability. 293T cells are well known for their rapid growth, ease of transfection, reproducibility and robust protein stability, allowing us to focus on the effects of the different ALK4 mutants. Nonetheless, we repeated Figure 5c in MDA-MB-231 LM2 cells and similar results were observed (Figure L2).

Figure L2: MDA-MB-231 LM2 cells were transfected with TβRII along with pcDNA3.1, WT ALK4, or DN ALK4. The cell surface level of TβRII and level of ALK4 and β-actin were assessed using western blotting.

Concern 14: Were any measures taken to ensure that observers were blinded to the experimental conditions, for example in the case of counting metastatic nodules or gross metastases in Fig2D-M. If so these should be described in the manuscript.

In general, wherever possible observers were blinded to the experimental conditions. For example, in the evaluation of cell migration and invasion, these were all quantitated by observers blinded to the experimental conditions. Similarly, for histology analysis, the observer (a Duke certified pathologist) was blinded to the experimental conditions. This practice is noted in the material and methods section.

Concern 15: Additional methodological detail should be provided. For example, in the TGFb treatment experiments, how was serum resting performed.

Throughout the manuscript, we have increased the level of methodological details, including details of serum starvation, in the corresponding figure legends.

Concern 16: What is the rationale for pre-treating for 72 hours with soluble activin receptor prior to 30min activin treatment? A shorter term incubation would be more appropriate.

We agree with the reviewer that for determining whether signaling mediator pSmad2 is changed, a short-term incubation would have been sufficient. However, we also wanted to investigate whether TGF- β signaling and TGF- β induced EMT were altered upon activin A inhibition, and given the high cost of the reagent (soluble activin receptor), we decided to perform a 3-day treatment to inhibit activin A in the serum throughout the 3 day experiment.

Concern 17: For Fig12H,I quantitation should be presented in bar graphs.

This quantitation is now shown in Supplemental Figure 15 (h-i).

Concern 18: Mgat5 mRNA levels should be measured in siMgat5 condition. The decrease in LPHA staining suggests poor inhibition of Mgat5. Or use Mgat5 KO as suggested above.

The high basal level of PHA-L stain in siMGAT5 condition is due to nonspecific background staining. We did perform RT-qPCR to validate the siMGAT5 knockdown condition and now include this data (sFigure 14b). In addition, we have performed the experiment with Mgat5 KO (new sFigure14c).

Concern 19: Quantitation of Fig S14F is needed.

Quantitative data for the lung lesions are shown in Figure 8m. We have removed sFigure 14F to reduce redundancy.

Reviewer #2

Reviewer #3 (Remarks to the Author):

We thank the reviewer for finding that the manuscript “presents extensive findings that support a highly novel concept,” and that this is “an extremely rigorous study that clearly presents highly innovative data that has the potential to lay the groundwork for a mechanistic understanding of EMT and metastasis not previously recognized.”

Concern 1: Several pieces of data are presented establishing that Alk4 is decreased in breast and PANC and that this is associated with disease progression. Some discussion

could be offered as to what events lead to the loss of Alk4, i.e. genetic, transcriptional, post-transcriptional.

We appreciate this concern. While ALK4 is mutated, there are other mechanisms contributing to loss of ALK4 function, including loss of copy number/loss of heterozygosity. In addition, we now provide evidence supporting epigenetic silencing of the ALK4 gene, including data demonstrating that treating pancreatic and breast cancer cells with 5-AZA increased ALK4 expression, and that methylation of the ALK4 promoter in tumor samples correlates with decreased ALK4 expression (new sFigure 2c-h). These data and an overall discussion of mechanisms of loss of ALK4 function have been added to the revised discussion.

Concern 2: NGI-1 is a highly insoluble compound. Some level of PD from the in vivo studies (i.e. changes in TBR-I or other protein glycosylation) would help support on-target effects of the observed anti-tumor activity.

We appreciate this concern. Indeed, during the in vivo analysis presented, we had the same concern and performed a thorough investigation to mitigate this challenge. As the PANC-1 NTC and ALK4 KO cells already carried the luciferase reporter system to assess for pulmonary lesions, we were not able to use the original luciferase based reporter system [5, 6] to assess for changes in glycosylation. Therefore, we investigated alternative approaches for drug delivery. We prepared the solution right before injection based on a recently published study [7]. In addition, to address the reviewer's concern, we performed IHC staining for PHA-L to support the effectiveness of NGI-1 treatment, demonstrating that NGI-1 decreased PHA-L levels. These data were have been added as new sFigure 18c.

Concern 3: The use of the term metastasis in reference to pulmonary tumors established by tail vein inoculation should be modified to pulmonary lesions or something similar.

We appreciate this concern. In response, for all tail vein studies, the description of the results has been changed to pulmonary lesions.

Reviewer #4

We thank the reviewer for their careful review of the manuscript.

Concern 1: While at a first glance the hypothesis that the loss of ALK4 contributes to cancer progression might seem appealing, the study does not provide compelling evidence to support such claim. It also lacks the conceptual advance needed for establishing the relevance of this mechanism to the human diseases. Indeed, the vast

majority of experiments are superficial in nature, relying mostly on in vitro cell systems, which could lead to all sorts of artifacts.

In this manuscript, we used different knockdown and knockout models including transient knockdown using siRNA, stable knockdown using shRNA, CRISPR knockout and Cre recombinase mediated genetic deletion to observe consistent effects of ALK4 loss on cancer progression. We further used unbiased proteomic analysis to establish mechanistic leads, and validated this mechanism in epistatic experiments, including in vivo studies. In addition to extensive in vitro analysis, we performed in vivo mouse studies including tail vein inoculation and orthotopically injection using 4 different cell lines with both loss and gain of function models, and now add a GEMM. We realize that any scientific method has its limitations, but utilizing multiple complementary approaches, we have provided consistent results.

Concern 2: The authors relied heavily on several public datasets to show that ALK4 is lost in breast and pancreatic cancers, with low levels of ALK4 correlating with poor prognosis. The authors might want to consider corroborating these data using qRT-PCR or immunohistochemistry with tissue microarrays. In any case, considering the possibility that ALK4 expression is lost in breast and pancreatic cancers, one would wonder what mechanisms could contribute to this phenomenon in the cancerous samples that do not display any mutation or loss of copy numbers? Is there any other genetic or epigenetic event that could compromise ALK4 expression? This information is important for understanding the exact contribution of ALK4 to cancer progression and metastasis.

We agree with the reviewer's concern. For the presented in silico analysis, we investigated in total 192 PAAD GEO datasets and 1,464 BRCA GEO datasets, the presented datasets meet the following criteria: 1) datasets that contain gene expression profile from normal tissue; and 2) datasets that contain a minimum of 3 samples in each group (normal tissue and tumor tissue). To corroborate our in silico data, as the reviewer suggested, we acquired RNA samples from patients with pancreatic primary tumors and adjacent normal tissue from Duke BioRepository & Precision Pathology Center. Consistent with published database, ACVR1B expression is significantly lower in tumor tissues (new Figure 1c).

While somatic mutation of ALK4 have been identified in a small percentage of breast and pancreatic cancers, there are other mechanisms contributing to loss of ALK4 function, including loss of copy number/loss of heterozygosity. In addition, we now provide evidence supporting epigenetic silencing of the ALK4 gene, including data demonstrating that treating pancreatic and breast cancer cells with the DNA methylation inhibitor 5-azacytidine (5-AZA) increased ALK4 expression, and that methylation of the ALK4 promoter in tumor samples

correlates with decreased ALK4 expression (new sFigure 2c-h). These data and an overall discussion of mechanisms of loss of ALK4 function have been added to the revised discussion.

Concern 3: The expression levels of ALK4 seem to be inversely correlated with the aggressiveness and metastatic potential of breast and pancreatic cancer cell lines. However, this correlation is based on very artificial invasion assays in vitro. To support their claim, the authors might want to consider characterizing these cell lines in vivo, both in terms of tumor growth and metastatic potential.

We appreciate this concern of the reviewer. To evaluate the metastatic potential of these cell lines, we performed tail vein inoculation using PANC-1, HPNE, Mia Paca 2 and ASPC1 cells expressing luciferase. Briefly, half million cells of each line were injected into 8 NSG mice and the average bioluminescence was detected every week for up to 6 weeks. As shown in Figure L3, we observed that ASPC1 was the most aggressive cells line, followed by Mia Paca 2 and then HPNE/PANC-1. Accordingly, we agree with the reviewer's concern and have deleted this correlative data from the revised manuscript.

Figure L3. Evaluation of metastatic potential of different PDAC cells. 5×10^5 ASPC1, Mia Paca2, HPNE, or PANC-1 cells were injected into the tail vein of NSG mice (n=8 per group). Pulmonary lesions were detected using in vivo imaging. (A) The fold change of bioluminescence of each group is shown. (B) Overall survival of mice injected with indicated PDAC cells.

Concern 4: To provide compelling evidence that the loss of ALK4 contributes to cancer progression, it is imperative to conduct in vivo experiments with autochthonous mouse models of breast and pancreatic cancer. It would be straightforward to delete AcvR1B in mouse models of breast and pancreatic cancer (such as in MMTV-PyMT or KC mice with conditional expression of KrasG12D in the pancreatic epithelium). These experimental approaches must be presented at his level of publication.

We agree with this concern, and in response, we have incorporated in vivo experiments with autochthonous mouse models of pancreatic cancer, with oncogenic KRAS^{G12D} driven by Pdx1-Cre, with or without ALK4 (ALK4^{fl/fl} loss), which demonstrated decreased survival and increased spontaneous metastases with loss of ALK4 (new sFigure 4). The corresponding GEMM mouse data is now described and discussed in the revised manuscript.

Concern 5: ALK4 seems to induce expression of T β RI and T β RII at the surface by a mechanism dependent on its kinase activity. What is the substrate of ALK4 under these conditions? This information is important for understanding how ALK4 fulfills its anti-tumor suppressive function in breast and pancreatic cancers. Also important is what signaling pathway functions downstream of ALK4 to suppress cancer progression and metastasis? Without providing this kind of data, the impact of the study will be just incremental.

We appreciate this concern. While we have not been able to define the precise substrate directly phosphorylated by ALK4 to mediate these effects, we have investigated the mechanism by which loss of ALK4 mediates its effects on increasing MGAT5 and galectin-3. As another reviewer suggested, we first have investigated whether non-canonical ALK signaling was affected after acute blockage of ALK4 via siRNA mediated silencing of ALK4 expression. As shown in new sFigure 16, siRNA mediated silencing of ALK4 expression increased p38 phosphorylation, without altering ERK or Akt signalling (sFigure 16a-c). However, inhibiting the p38 pathway using the small molecule inhibitor SB203580 did not affect TGF- β signaling or expression of MGAT5/galectin-3 (sFigure 16d-f). We also investigated whether ALK4 loss mediated increases in TGF- β signaling were responsible for upregulating Mgat5/galectin-3 to induce expression of T β RI and T β RII, and demonstrated that they were not (sFigure 15). Finally, we investigated whether ALK4 loss upregulates Mgat5/galectin-3 through a specific transcription factor. As shown in new sFigure 17, we found that ETS1 expression is significantly increased by ALK4 loss, and that silencing of ETS1 decreased MGAT5 induction mediated by ALK4 loss expression, supporting a role of ETS1 downstream of ALK4 loss.

Concern 6: Loss of ALK4 correlates with increased expression of MGAT5 and galectin-3, which could provide a mechanistic explanation as to how ALK4 regulates the N-glycosylation and thereby stability of T β RI and T β RII at the cell surface. There are at least three issues with this observation. First, the mechanism leading to the accumulation MGAT5 and galectin-3 in cells deficient for Alk4 was not explored. Second, compelling evidence that T β RI and T β RII are direct substrate of MGAT5 and galectin-3 was not presented. Third, compelling evidence that MGAT5 and galectin-3 functions downstream of ALK4 in vivo under physiological conditions is missing. The latter could be easily achieved through using mice or cells co-deleted of MGAT5 /galectin-3 and Alk4 and treated with the two inhibitors NGI-1 and Glucosamine.

We have investigated the mechanism by which loss of ALK4 mediates its effects on increasing MGAT5 and galectin-3, as addressed in Concern 5. In terms of whether T β RI and T β RII are direct substrates of MGAT5 and galectin-3, this connection was established by

Partridge et al [8], so we did not re-address here. We did establish that MGAT5 modified branched N- glycan were increased on T β RII (Figure 7d) and that internalization of T β RII level is decreased in the absence of ALK4 (Figure 7g) to further support this connection. To address whether MGAT5 is downstream of ALK4, we developed isogenic MGAT5 CRISPR KO cells using the PANC-1 NTC and ALK4 KO cells respectively. The knockout of MAGT5 was confirmed using flow cytometry (new sFigure 14c). As shown in new Figure 7h-k, we observed a significant decrease in anchorage-independent growth and development of pulmonary lesions when MGAT5 is depleted in ALK4 KO cancer models, supporting a role for MGAT5 mediated branched N-glycan modifications in mediating the effects downstream of ALK4 loss.

Concern 7: Overall, the study lacks the mechanistic depth for firmly establishing a role for ALK4 in suppressing cancer progression and metastasis. It also lacks the mechanistic depth for firmly establishing how ALK4 regulates the stability of T β RI and T β RII at the cell surface and attendant activation of TGF- β /Smad signaling pathway. Expression of ALK4 at the cell surface might simply rely on the same mechanism as does localization of T β RI and T β RII, so one conceivable mechanism would be that ALK4 might compete with T β RI and T β RII for the machinery that drives their cell surface localization.

We believe we have presented strong evidence for ALK4 in suppressing cancer progression, and mechanistic insight into how ALK4 regulates the stability of T β RI and T β RII, via MGAT5 and galectin-3. In terms of whether ALK4 might compete with T β RI and T β RII for the machinery that drives their cell surface localization, given the similarity between ALK4 and T β RI we considered that possibility, however our proteomic data and preliminary data support the upregulation of a broad spectrum of glycosylated cell surface proteins/receptors, including integrins in PANC-1 cells and VEGFR2 in endothelial cells (Figure 6g and Figure L4), making this less likely. This is now discussed.

Figure L4. Loss of ALK4 promotes VEGFA-VEGFR2 signaling through regulating VEGFR2 glycosylation and stability. (A) MEEC cells expressing shNTC or shALK4 were treated with VEGF-A (50ng/ml) for the indicated times, and cell surface levels of VEGFR2 were assessed by a cell surface biotinylation assay. (B) Beads were coated with PHA-L lectin and incubated with protein isolated from MEEC NTC and ALK4 KD cell lysate. Eluted samples and whole cell lysate were analyzed for VEGFR2. (C) MEEC cell expressing shNTC and shALK4 were treated with or without VEGFA for 10min. Protein lysate from MEEC shNTC and shALK4 cells were analyzed for indicated protein using western blotting.

Reviewer #5

We thank the reviewer for finding that the manuscript presents three major findings, each “supported individually by a large body of data”

Concern 1: The mechanism that links loss of ALK4 function to Gal-3 and Mgat5 transcription and thereby glycosylation of TGF-beta receptors is missing.

We appreciate this concern. In response we investigated whether non-canonical TGF- β signaling was affected after acute blockage of ALK4 via siRNA mediated silencing of ALK4 expression. As shown in new sFigure 16, siRNA mediated silencing of ALK4 expression increased p38 phosphorylation, without altering ERK or Akt signaling (sFigure 16a-c). However, inhibiting the p38 pathway using the small molecule inhibitor SB203580 did not affect TGF- β signaling or expression of MGAT5/galectin-3 (sFigure 16d-f). We also investigated whether ALK4 loss mediated increases in TGF- β signaling were responsible for upregulating Mgat5/galectin-3, and demonstrated that they were not (sFigure 15). Finally, we investigated whether ALK4 loss upregulates Mgat5/galectin-3 through a specific transcription factor. As shown in new sFigure 17, we found that ETS1 expression is significantly increased by ALK4 loss, and that silencing of ETS1 decreased MGAT5 induction mediated by ALK4 loss expression, supporting a role of ETS1 downstream of ALK4 loss.

Concern 2: The data/experiment description are insufficient, imprecise/inaccurate, and often hard to follow.

We have worked with a professional scientific writer/editor to improve the detail and clarity of the description of data and experiments in the results, methods, and figure legends.

Concern 3: Comparison of ALK4 levels in various cancer cell lines is not meaningful or superficial at best (sFig.2C-D), as underlying causal factors vary from one cell type to another. For the same reason, comparison in sFig.9A is also problematic, e.g. ALK4 level is similar between 168FARN and 4T07 (sFig 2D) but cell surface TGFbR are quite different (sFig 9A). Moreover, the data were too scarce to support the correlation.

We appreciate this concern of the reviewer. To evaluate the metastatic potential of these cell lines, we performed tail vein inoculation using PANC-1, HPNE, Mia Paca 2 and ASPC1 cells expressing luciferase. Briefly, half million cells of each line were injected into 8 NSG mice and the average bioluminescence was detected every week for up to 6 weeks. As shown in Figure L3, we observed that ASPC1 was the most aggressive cell line, followed by Mia Paca 2 and then HPNE/PANC-1. Accordingly, we agree with the reviewer’s concern and have deleted this correlative data from the revised manuscript.

Concern 4: Authors made several lines of stable cells expressing shALK4 or ALK4 gRNA. These lines were used interchanged throughout the manuscript. It is not clear why some experiments using the shALK4 lines while others using the ALK4KO lines. Sometimes panels were labelled with ALK4KO while legend said otherwise.

This is a study that has been going on for several years. The lab first started the investigation of ALK4 using cells stably expressing shRNA. With the emergence of CRISPR-Cas9 technique, we then incorporated the CRISPR KO system into our study as the main approach to stably knockout ALK4 due to higher fidelity and efficiency. Therefore, different cell models were presented in this manuscript. The presented phenotypical evaluation has been tested using both knockdown and knockout models. We have carefully examined the figures and corresponding legends to make sure it is clear which model is being used in each study.

Concern 5: Experiments in (Fig.2G-M were inadequately described. Legend 2G says the cells were implanted in “the tail of pancreas”. Were they implanted in the pancreas or injected in tail vein?

These experiments (now Figure 2 g-k) are orthotopic injection into the tail of mouse pancreas, as recommended by the previously published protocols [9, 10]. Supplemental Figure 3 (c-i) are orthotopic injection to mouse 4th mammary gland. Both analyses support that ALK4 suppresses spontaneous metastasis in these orthotopic models. This is now clearly indicated in the manuscript.

Concern 6: It is known that both activin and TGF- β are capable of inducing Smad2 and Smad3 phosphorylation. And it appears that ALK4 KO by itself consistently induced Smad2 phosphorylation, and enhanced TGF- β -mediated Smad2/3 phosphorylation (Fig 4A-C). Again, the legend in Fig 4C was different from the panel label. These experiments were done in cancer cells, phosphorylation of Smad2/3 should also be examined in cancer models.

We have corrected the figure legend for Figure 4c. We also examined pSmad2 in mouse pancreas injected with HPNE shNTC and shALK4 cells as shown in Supplemental Figure 7D.

Concern 7: “Clinically” is inappropriately used in describing results under “loss of ALK4 induces EMT...”, as the finding described was from analysis of human samples, not clinical studies. Fig.4E is poorly described and not clear what the conclusion was reached, and it is certainly not a dot-plot.

We have corrected the description as patient data/in silico data. Also Figure 4e has been rephrased to accurately reflect the data.

Concern 8: Fig. 4H, why DN-TbRII did not block TGF- β induction of Smad2 phosphorylation (shNTC data group)?

The original Figure 4H was performed with a 2 hour TGF- β treatment while all other WB assessing pSmad2 were performed with a 30 min TGF- β 1 treatment. Therefore, we repeated this study with 30min TGF- β 1 treatment 3 times and a representative figure panel and quantification of pSmad2/tSmad2 is shown as new Figure 4H. The new control demonstrating successful transfection of DN-RII is shown in sFigure 8a.

Concern 9: The sActRIIB-Fc experiments appear to show that effects of ALK4 KO on TGF- β signaling not due to activin signaling per se. Since ALK4 was once thought to mediate TGF- β signaling and it does possess some effect, could it be that there is a negative feedback loop connecting ALK4 to TGF- β signaling?

While we have considered the possibility that there is negative feedback loop connecting ALK4 to TGF- β signaling, given that loss of ALK4 function increases the levels of multiple cell surface receptors beyond TGF- β receptors, and their downstream signaling, we believe that the delineated pathway is the most likely explanation for the identified effects. This is now discussed in the revised manuscript.

Concern 10: Fig.5A lacks controls showing ALK4 levels after KO. Fig.5D missing ALK4 expression control. sFig.9E is a haphazard control for Fig.5D.

The ALK4 KD control for Fig. 5a is shown in new Supplemental Figure 11a. The ALK4 OE control for Figure 5d is shown in Supplemental Figure 3b.

Concern 11: Galectin-3 was singled out based on mapped interactions with TGF- β targeted genes, coreceptors, FN1, MMP2, CD44, and endoglin in the STRING analysis. However, there are quite a few other nodes in this interaction network that have not been examined or ruled out. No compelling reason was given to proceed with Gal-3. It appears to be cherry picked.

While STRING analysis was used to identify potential interactors, we ranked all clustered genes based on the number of interactions. The top 5 genes ranked in sequences were: FN1 (19), ITGB3 (14), THBS1 (13), LGALS3 (11), and CD44 (10). All of these genes, with the exception of galectin-3, were known to be regulated by TGF-beta signaling. In addition, many of the other genes linked to galectin-3 were TGF-beta regulated genes, thus our focus on galectin-3. We didn't ignore other important nodes like integrins (ITGA3, ITGB3, ITGB4), and we are now writing another manuscript focusing on ALK4 regulation of integrins in pancreatic cancer. We have now described the rationale for focusing on galectin-3 in the results section.

Concern 12: More importantly, authors showed that knockdown or KO ALK4 increased Gal-3 and Mgat5 transcription, the increased Gal-3 and Mgat5 promoted glycosylation of TGF- β receptors and increased TGF- β signaling, however, siTbRI had no effect on Gal-3 and Mgat5 transcription (sFig 12). Then how did shALK4 (or ALK4KO) enhance Gal-3 and Mgat5 transcription? Did Smad2 or Smad3 play a role in their transcription?

We have tested the role of Smad2 by knocking down Smad2 with specific siRNA (Supplemental Figure 15j-k), demonstrating that induction of galectin-3 and MGAT5 was not changed. However, we did find that loss of ALK4, at least partially, upregulates MGAT5 expression through the ETS1 transcription factor (new Supplemental Figure 17).

References

1. Zhang, H., et al., *Common variants in glutamine:fructose-6-phosphate amidotransferase 2 (GFPT2) gene are associated with type 2 diabetes, diabetic nephropathy, and increased GFPT2 mRNA levels.* J Clin Endocrinol Metab, 2004. **89**(2): p. 748-55.
2. Lau, K.S., et al., *Complex N-glycan number and degree of branching cooperate to regulate cell proliferation and differentiation.* Cell, 2007. **129**(1): p. 123-34.
3. Nielsen, M.I., et al., *Galectin binding to cells and glycoproteins with genetically modified glycosylation reveals galectin-glycan specificities in a natural context.* J Biol Chem, 2018. **293**(52): p. 20249-20262.
4. Hirabayashi, J., et al., *Oligosaccharide specificity of galectins: a search by frontal affinity chromatography.* Biochim Biophys Acta, 2002. **1572**(2-3): p. 232-54.
5. Contessa, J.N., et al., *Molecular imaging of N-linked glycosylation suggests glycan biosynthesis is a novel target for cancer therapy.* Clin Cancer Res, 2010. **16**(12): p. 3205-14.
6. Baro, M., et al., *Oligosaccharyltransferase Inhibition Reduces Receptor Tyrosine Kinase Activation and Enhances Glioma Radiosensitivity.* Clin Cancer Res, 2019. **25**(2): p. 784-795.
7. Scheich, S., et al., *Targeting N-linked Glycosylation for the Therapy of Aggressive Lymphomas.* Cancer Discov, 2023. **13**(8): p. 1862-1883.
8. Partridge, E.A., et al., *Regulation of cytokine receptors by Golgi N-glycan processing and endocytosis.* Science, 2004. **306**(5693): p. 120-4.
9. Jiang, Y.J., et al., *Establishment of an orthotopic pancreatic cancer mouse model: cells suspended and injected in Matrigel.* World J Gastroenterol, 2014. **20**(28): p. 9476-85.
10. Pang, T.C.Y., et al., *An Orthotopic Resectional Mouse Model of Pancreatic Cancer.* J Vis Exp, 2020(163).

Point-by-point response

Reviewer #1

We thank the reviewer for finding that the original manuscript is “interesting and thoroughly investigated” and provides “compelling evidence” supporting its central claims, including that the “novelty in this manuscript” lies in the connection of ALK4 down regulation to Mgat5/galectin-3 up-regulation. For the revised manuscript, we thank the reviewer for finding that “The authors have made a solid attempt to address our concerns.”

Concern 1: Recommend to put ETS1 data in main figure, and cite known regulation of Mgat5 by ETS1

This has been addressed, ETS1 data are now included in new Figure 8, and regulation of MGAT5 by ETS1 is cited in the revised manuscript (lines 348-349).

Concern 2: the new Mgat5KO data is convincing, but recommend that TGF β signaling be assessed as in Fig8A. Also again strongly recommend that Fig. 8 should be moved to supplemental figures and the new Mgat5KO data be its own Figure.

The recommended evaluation of TGF- β signaling has been performed and is now shown as new sFig. 14D. We have also moved original Figure 8 to Supplementary figures (sFig. 18 and 19) as the reviewer recommended.

Concern 3: The spontaneous metastatic data is limited (see issues below) and at the very least the interpretation should acknowledge that the effects on metastasis were less clear.

In Fig. 2k - Statistical analysis for this panel is confusing. 3/8 is 37.5% not 45%. The figure legend states that this was done by Mann-Whitney. Statistical analysis should be re-done as a contingency table and clarifying whether the same mouse had multiple mets or all mice with mets were different mice. Alternatively, the number of mets per mouse could be shown leaving aside location.

We reformatted the panel and performed the statistical analysis as recommended. The updated p values are labeled in the figure, and the statistical analysis is updated in figure legends. These mice all have either an intrasplenic, mesenteric/diaphragm or liver metastasis, with no mouse having multiple sites of metastasis. This is now clarified in the revised manuscript (lines 531-532).

In Fig. S3g, lung mets should be quantitated.

The lung metastases were quantified by in vivo imaging in sFig. 3f and sFig. 3h. sFig.3g is representative images of the development of lung metastatic lesions.

In Fig. S4, ascites can reflect local invasion rather than metastasis and the two should not be grouped together.

Thank you, we agree, and this table has been revised based on both reviewer’s comments accordingly.

sFIG4a: x-axis should say days not months

Thank you, this has been corrected.

Concern 4: Please clarify the lectin array methodology in the manuscript. Is this using lysate or cells?

The lectin array was done using cell lysate. This is now specified in the methods section (lines 642-646).

Figure FigL1 should be included in the paper with a clear definition of the mass spec assay used. Is this UDP-HexNAc or UDP-GlcNAc? Discuss discrepancy between UDP-GlcNAc and lectin array. Why so few changes in lectin binding? Also, why no increases in Galectin staining as expected?

(1) **“Figure FigL1 should be included in the paper with a clear definition of the mass spec assay used. Is this UDP-HexNAc or UDP-GlcNAc?”**

This data has now been added as new sFig. 13 d-g. The mass spec assay to detect levels of UDP-GlcNAc was developed by Duke Proteomics and Metabolomics Core Facility. Briefly, PANC-1 crNTC and crALK4 KO cells and MDA-MB-231 crNTC and crALK4 KO cells (3 million each, 4 replicates for each condition) were lysed and incubated with 13C6- UDP-GlcNAc (Omicron Biochemicals, Inc.). Samples were analyzed with a multiple reaction monitoring (MRM) assay. Data was acquired using UPLC-MS/MS on a Sciex QTrap 6500+ mass spectrometer, including calibration curves for the analyte in a manner consistent with the FDA Guidance for Bioanalytical Method Validation. Data analysis was done using Skyline. The methodology is described in methods section. (lines 647-656)

(2) **“Discuss discrepancy between UDP-GlcNAc and lectin array.”** and question **“Why so few changes in lectin binding?”**

Previously published reports have shown that the MGAT5 Km value for UDP-GlcNAc is considerably higher compared to that observed in other N-acetylglucosaminyltransferases (e.g. MGAT1, MGAT2, and MGAT4) [1, 2]. As for the lectin array, there are 7 lectins with significantly changed staining, but only two with a significant 2-fold change (increase).

(3) **“Also, why no increases in Galectin staining as expected?”**

The intensity of galectin-3 stain is slightly increased in KOs compared to control group, while this change is not significant. Galectin-3 recognizes poly LacNAc while PHA-L recognize Gal β 4GlcNAc β 6(GlcNAc β 2MAN α 3)MAN α 3. Considering that ALK4 loss specifically increased MGAT5-modified branched N-glycan, likely the change is “diluted” by other glycans competing for binding to galectin-3.

Concern 5: The authors have largely updated the abstract, introduction and discussion to better reflect the published data that N-glycan branching, but a sentence in the abstract should be changed (as highlighted in red) below to be more accurate:

“Consistent with prior observation that galectin-3 preferentially binds to MGAT5-modified glycoproteins to stabilize cell surface receptors like TGF- β receptors, we found that ALK4 loss enhanced MGAT5-mediated glycosylation of TGF- β receptors...”

This change has been made to the abstract.

Concern 6-8: Previously addressed.

Concern 9: Some figure legends state that in vitro experiments were done with 3 biological replicates, but do not state if the experiments were independently replicated. This should be clarified. Some figure legends state the number of independent experiments but others do not. This should be clearly stated in all figure legends.

All in vitro experiments were performed with at least 3 independent biological replicates, this is now clearly stated in all figure legends.

Concern 10: Some WBs have been quantitated in the figures, others have been presented as a supplemental excel file. It would be best if all WB quantitation was present within the figures (at least the main figures).

We sincerely appreciate the reviewer's insightful suggestion regarding the presentation of Western blot quantitation. In accordance with this recommendation, we have carefully re-evaluated our data presentation strategy to optimize clarity while maintaining scientific rigor.

The quantitative analyses of comparative experiments evaluate Smads phosphorylation or TβRs level which need comparison between 2 blots (Figures 4a,b,c,h; 7d,e,f,g) and time-course studies (Figure 5a,b,e,i) have been retained in the main figures to increase clarity. For panels demonstrating straightforward binary relationships (e.g., Figure 3a,b, Figure 7a) or mobility shifts (e.g., Figure 5c,d,f,g,h), we have opted to present these datasets in the supplementary Excel file. This decision was guided by our concern that superimposing numerical quantitations on such panels might compromise visual clarity. For example, as shown in the figure below, we think that the previous unlabeled version (right) is more straightforward. To ensure full transparency, we described our strategy in methods section (lines 606-608). This balanced approach aims to maintain narrative focus while providing complete data accessibility.

Concern 13: Figure L2 should be in the manuscript in the main figure as Fig5d.

This change has been incorporated into the revised manuscript.

Concern 14: We could not find blinding in the methods section, this should be corrected. In addition, the figure legends should state when blinding was performed.

This is now mentioned in material and methods section (line 543, line 551, line 569, line 591) and describe in the figure legends.

Other Minor concerns:

Figure 1 legend is incomplete or cut off in the PDF.

Figure 1 legend was cut off and we have fixed this issue.

The figures often refer only to “KO”. These should be fixed to identify the specific KO (eg “ALK4 KO”)

This has been performed across all figures to increase clarity.

Reviewer #2 (co-reviewer of Reviewer #1, withdrawn)

Reviewer #3 (Remarks to the Author):

We thank the reviewer for finding that the original manuscript “presents extensive findings that support a highly novel concept,” and that this is “an extremely rigorous study that clearly presents highly innovative data that has the potential to lay the groundwork for a mechanistic understanding of EMT and metastasis not previously recognized.”

We thank the reviewer for finding that in the revised manuscript “The authors have addressed my previous, minor, critiques. The authors should be commended on completion of this highly impactful, rigorous, and scientifically interesting manuscript.”

Abstract line 44; consistent with prior observations

This has been corrected.

Reviewer #4

We thank the reviewer for their careful review of the revised manuscript, for noting that “The observation that ALK4 may function as a tumor suppressor in breast and pancreatic cancers by antagonizing T β RI/T β RII is intriguing and warrants further mechanistic and translational studies” and that the “authors have made significant efforts to address the concerns raised by both this reviewer and others.”

Reviewer concern 1: While at a first glance the hypothesis that the loss of ALK4 contributes to cancer progression might seem appealing, the study does not provide compelling evidence to support such claim. It also lacks the conceptual advance needed for establishing the relevance of this mechanism to the human diseases. Indeed, the vast majority of experiments are superficial in nature, relying mostly on in vitro cell systems, which could lead to all sorts of artifacts.

Author response: In this manuscript, we used different knockdown and knockout models including transient knockdown using siRNA, stable knockdown using shRNA, CRISPR knockout and Cre recombinase mediated genetic deletion to observe consistent effects of ALK4 loss on cancer progression. We further used unbiased proteomic analysis to establish mechanistic leads, and validated this mechanism in epistatic experiments, including in vivo studies. In addition to extensive in vitro analysis, we performed in vivo mouse studies including tail vein inoculation and orthotopically injection using 4 different cell lines with both loss and gain of function models and now add a GEMM. We realize that any scientific method has its limitations, but utilizing multiple complementary approaches, we have provided consistent results.

New Reviewer critique: The authors have made significant efforts to address the concerns raised by both this reviewer and others. However, compelling evidence supporting ALK4's role as a tumor suppressor in breast and pancreatic cancers remains insufficient.

Numerous advanced technologies could be leveraged to elevate this potentially intriguing study, providing the depth needed to justify its publication in a high-profile journal.

We appreciate the reviewer's rigorous evaluation of our work. Respectfully, we would like to emphasize that our comprehensive approach integrating patient data analysis demonstrating ALK4's prognostic significance (Figure 1 and Supplementary Figure 1), functional characterization using isogenic cell models (generated using shRNA or CRISPR), xenograft tumorigenicity assays (Figure 2, 7 and Supplementary Figure 3, 18, 19), genetically engineered mouse models (Supplementary Figure 4), and unbiased proteomic analysis for mechanistic investigation (Figure 6), collectively provides validation across human populations, cellular systems, and in vivo mammalian models. We have worked hard to ensure the accuracy and fidelity of our work, and acknowledge that there is always more that could be done.

Reviewer concern 2 has been resolved.

Reviewer concern 3: The expression levels of ALK4 seem to be inversely correlated with the aggressiveness and metastatic potential of breast and pancreatic cancer cell lines. However, this correlation is based on very artificial invasion assays in vitro. To support their claim, the authors might want to consider characterizing these cell lines in vivo, both in terms of tumor growth and metastatic potential.

Author response: We appreciate this concern of the reviewer. To evaluate the metastatic potential of these cell lines, we performed tail vein inoculation using PANC-1, HPNE, Mia Paca 2 and ASPC1 cells expressing luciferase. Briefly, half million cells of each line were injected into 8 NSG mice and the average bioluminescence was detected every week for up to 6 weeks. As shown in Figure L3, we observed that ASPC1 was the most aggressive cells line, followed by Mia Paca 2 and then HPNE/PANC-1. Accordingly, we agree with the reviewer's concern and have deleted this correlative data from the revised manuscript.

New Reviewer critique: The authors have not addressed this concern, which is crucial for demonstrating that ALK4 loss is associated with metastasis. Given that they have already conducted in vivo experiments using cell lines with varying ALK4 expression levels, it should be relatively straightforward to evaluate their metastatic potential by analyzing organs from the mice that received these cells (already done). Failing to do so could undermine the validity of their working model.

As Reviewer #5 commented that "Comparison of ALK4 levels in various cancer cell lines is not meaningful or superficial at best, as underlying causal factors vary from one cell type to another". Indeed, these cells line were derived from different origins with different molecular profiles, as shown in the table below.

	Derivation	Gender	Kras	TP53	CDKN2 A	SMAD 4	TGFBR II
ASPC1	ascites [3]	Female	12 Asp[4, 5]	135 Δ 1 bp[5, 6]	Δ 2 bp[5, 7]	HD[8]	N/A
Mia Paca-2	pancreatic tumor	Male	12 Cys[5]	248 Trp[5]	HD[5]	WT[5]	NE[9]

PANC-1	pancreatic tumor [10]	Male	12 Asp[4]	273 His/Cys[4, 11]	HD[4]	WT[4]	WT[12]
--------	-----------------------	------	-----------	--------------------	-------	-------	--------

HD: homozygous deleted; N/A: not available; WT: wild-type; NE: not expressed

In addition, we also investigated the comparison of in vivo tumorigenicity using different pancreatic cancer cell lines with opposing data observed [13-15]. In two studies, BxPC-3 tumors were consistently larger than PANC-1 tumors whereas a third showed the opposite. Therefore, the conclusion from these studies would not be as valid as altering ALK4 expression in the same genetic background using cell, xenograft, and GEM model and evaluating the cancer progression as already performed.

Reviewer concern 4: To provide compelling evidence that the loss of ALK4 contributes to cancer progression, it is imperative to conduct in vivo experiments with autochthonous mouse models of breast and pancreatic cancer. It would be straightforward to delete AcvR1B in mouse models of breast and pancreatic cancer (such as in MMTV-PyMT or KC mice with conditional expression of KrasG12D in the pancreatic epithelium). These experimental approaches must be presented at his level of publication.

Author response: We agree with this concern, and in response, we have incorporated in vivo experiments with autochthonous mouse models of pancreatic cancer, with oncogenic KRASG12D driven by Pdx1-Cre, with or without ALK4 (ALK4^{fl/fl} loss), which demonstrated decreased survival and increased spontaneous metastases with loss of ALK4 (new sFigure 4). The corresponding GEMM mouse data is now described and discussed in the revised manuscript.

New Reviewer critique: The authors have not addressed whether ALK4 ablation in the MMTV-PYMT mouse model (or an equivalent model) suppresses mammary tumor progression and metastasis. Given that they already have floxed ALK4 mice, this experiment should be relatively straightforward to conduct. However, the authors conducted experiments using the KC mouse model of pancreatic cancer and demonstrated that Alk4 deletion in the KrasG12D background accelerates pancreatic cancer progression and metastasis. They should quantify the H&E staining for all examined pancreatic tissues and provide at least a histological analysis of the organs affected by metastasis (e.g., liver, lung, etc.). Additionally, to further validate their working model, they should present evidence of increased TGF β signaling, elevated expression of MGAT5, and upregulated T β RI and T β RII expression, along with enhanced N-glycan modifications of these receptors. This is the most robust model system presented in the study, and they should fully leverage its potential.

We thank the reviewer for the thoughtful suggestion regarding the evaluation of ALK4 ablation in the MMTV-PYMT mouse model. We agree that such an experiment could provide valuable insights into the role of ALK4 in mammary tumor progression and metastasis. However, conducting this study would require the generation and characterization of a new cohort of compound transgenic mice, as well as extensive in vivo analysis that falls outside the current scope and timeline of this study. Our current work describing a novel role of ALK4 in regulating TGF- β signaling and promoting cancer progression in established tumor/cell populations, using complementary in vitro and in vivo models. We believe that the findings presented here provide

a solid foundation and strong rationale for future in vivo studies focused on the role of ALK4 in tumor progression and metastasis in breast cancer. We have added a statement to the Discussion section to acknowledge this limitation and to highlight the importance of future in vivo work using autochthonous mouse models to fully elucidate the functional relevance of ALK4 in breast cancer biology (lines 423-426).

We have validated our working model in the KC GEM model as recommended. In addition, as recommended, we have examined pancreatic tissue from 10 KC and 10 AKC aged matched mice and among these, all AKC mice developed PDAC while only 3/10 KC mice developed PDAC (p value= 0.0031). The rest of KC mice developed low/high grade of PanIn lesion. This data has been presented in Supplemental Figure 4. A summary of the histological evaluation performed by a board certificate pathologist is provided as new Table 2. We also updated sFig. 4b to focus on ascites and the development PDAC, as there is was no significant difference in the number of metastatic lesions in AKC compared to KC. In response to the reviewer's third point, we further validated our working model using GEM mouse model. As shown in new sFig. 17, depletion of ALK4 in KC mouse pancreas significantly increased expression of galectin-3, MGAT5, ETS1 and TGF- β regulated genes which are known regulators/mediators of EMT.

Reviewer concern 5: ALK4 seems to induce expression of T β RI and T β RII at the surface by a mechanism dependent on its kinase activity. What is the substrate of ALK4 under these conditions? This information is important for understanding how ALK4 fulfills its anti- tumor suppressive function in breast and pancreatic cancers. Also important is what signaling pathway functions downstream of ALK4 to suppress cancer progression and metastasis? Without providing this kind of data, the impact of the study will be just incremental.

Author response: We appreciate this concern. While we have not been able to define the precise substrate directly phosphorylated by ALK4 to mediate these effects, we have investigated the mechanism by which loss of ALK4 mediates its effects on increasing MGAT5 and galectin-3. As another reviewer suggested, we first have investigated whether non-canonical ALK signaling was affected after acute blockage of ALK4 via siRNA mediated silencing of ALK4 expression. As shown in new sFigure 16, siRNA mediated silencing of ALK4 expression increased p38 phosphorylation, without altering ERK or Akt signaling (sFigure 16a-c). However, inhibiting the p38 pathway using the small molecule inhibitor SB203580 did not affect TGF- β signaling or expression of MGAT5/galectin-3 (sFigure 16d-f). We also investigated whether ALK4 loss mediated increases in TGF- β signaling were responsible for upregulating Mgat5/galectin-3 to induce expression of T β RI and T β RII and demonstrated that they were not (sFigure 15). Finally, we investigated whether ALK4 loss upregulates Mgat5/galectin-3 through a specific transcription factor. As shown in new sFigure 17, we found that ETS1 expression is significantly increased by ALK4 loss, and that silencing of ETS1 decreased MGAT5 induction mediated by ALK4 loss expression, supporting a role of ETS1 downstream of ALK4 loss.

New Reviewer critique: The response to this issue is insubstantial and fails to provide convincing evidence that ETS1 is indeed the transcription factor regulating MGAT5 expression. While existing literature supports the role of ETS1 in regulating MGAT5 expression, the authors have not investigated how ALK4 loss leads to increased ETS1 expression. Consequently, the approach taken to address this concern remains superficial.

The authors could strengthen their mechanistic insights by leveraging their KC and AKC mouse models to perform omics analyses (e.g., RNA-Seq, ATAC-Seq, spatial transcriptomics) to establish a more robust mechanistic framework, particularly regarding the interactions between ALK4, MGAT5, T β RI, T β RII, and ETS1.

We thank the reviewer for raising this important point regarding the mechanistic link between ALK4 loss, ETS1 expression, and MGAT5 regulation. To address this within the scope of the current study, we utilized our established KC and AKC genetically engineered mouse models to assess the expression of ETS1, MGAT5, and multiple TGF- β target genes in mouse pancreas by RT-qPCR. These analyses revealed coordinated upregulation of ETS1 and MGAT5 in the absence of ALK4, along with elevated expression of canonical TGF- β -responsive genes, supporting the proposed pathway in this GEMM (new sFig. 17). While we recognize that comprehensive omics approaches could further enhance mechanistic resolution, such analyses are beyond the scope and design of the current study. Instead, we focused on a targeted, hypothesis-driven strategy using in vitro and in vivo models that directly reflect ALK4 loss in a physiologically relevant context. We have clarified this point in the revised Discussion section and emphasized that future unbiased omics-based analyses will be valuable for deepening our understanding of ALK4-dependent transcriptional programs in pancreatic cancer (lines 454-457).

Reviewer concern 6: Loss of ALK4 correlates with increased expression of MGAT5 and galectin-3, which could provide a mechanistic explanation as to how ALK4 regulates the N-glycosylation and thereby stability of T β RI and T β RII at the cell surface. There are at least three issues with this observation. First, the mechanism leading to the accumulation MGAT5 and galectin-3 in cells deficient for Alk4 was not explored. Second, compelling evidence that T β RI and T β RII are direct substrate of MGAT5 and galectin-3 was not presented. Third, compelling evidence that MGAT5 and galectin-3 functions downstream of ALK4 in vivo under physiological conditions is missing. The latter could be easily achieved through using mice or cells co-deleted of MGAT5 /galectin-3 and Alk4 and treated with the two inhibitors NGI-1 and Glucosamine.

Author response: We have investigated the mechanism by which loss of ALK4 mediates its effects on increasing MGAT5 and galectin-3, as addressed in Concern 5. In terms of whether T β RI and T β RII are direct substrates of MGAT5 and galectin-3, this connection was established by Partridge et al [8], so we did not re-address here. We did establish that MGAT5 modified branched N- glycan were increased on T β RII (Figure 7d) and that internalization of T β RII level is decreased in the absence of ALK4 (Figure 7g) to further support this connection. To address whether MGAT5 is downstream of ALK4, we developed isogenic MGAT5 CRISPR KO cells using the PANC-1 NTC and ALK4 KO cells respectively. The knockout of MAGT5 was confirmed using flow cytometry (new sFigure 14c). As shown in new Figure 7h-k, we observed a significant decrease in anchorage-independent growth and development of pulmonary lesions when MGAT5 is depleted in ALK4 KO cancer models, supporting a role for MGAT5 mediated branched N-glycan modifications in mediating the effects downstream of ALK4 loss.

New Reviewer critique: This concern was addressed only partially and indirectly. To rigorously resolve this issue, the authors should reconstitute MGAT5-deleted cells with a catalytically inactive MGAT5 mutant that is incapable of mediating branched N-glycan modifications on T β RI/T β RII and demonstrate that this mutant fails to rescue the

anchorage-independent growth and metastatic phenotypes. Without this approach, it would be challenging to definitively establish that MGAT5 functions through its ability to mediate branched N-glycan modifications.

It has been previously established that MGAT5 functions through catalyzing β 1,6-branched N-glycans [16, 17], and numerous studies have demonstrated that this modification critically regulates cell surface receptor retention and signaling, including TGF- β receptors [17, 18] and promotes cancer progression and metastasis [18-22]. Expression of Mgat5 does rescue the cell surface receptor responsiveness and enhance the metastatic phenotype when the cells were injected into mice [18], while expression of the Mgat5 (L188R) mutation, which blocks enzyme localization to Golgi [23], failed to rescue signaling and EMT indicating a requirement for the N-glycan product of the enzyme [18]. Given this established enzymatic role, combined with our functional data showing that MGAT5 deletion using siRNA and gRNA that target different genetic locations impairs TGF- β signaling, cell migration, anchorage-independent growth and lung lesions, we believe the current evidence strongly supports the conclusion that MGAT5 promotes these phenotypes.

Reviewer concern 7: Overall, the study lacks the mechanistic depth for firmly establishing a role for ALK4 in suppressing cancer progression and metastasis. It also lacks the mechanistic depth for firmly establishing how ALK4 regulates the stability of T β RI and T β RII at the cell surface and attendant activation of TGF- β /Smad signaling pathway. Expression of ALK4 at the cell surface might simply rely on the same mechanism as does localization of T β RI and T β RII, so one conceivable mechanism would be that ALK4 might compete with T β RI and T β RII for the machinery that drives their cell surface localization.

Author response: We believe we have presented strong evidence for ALK4 in suppressing cancer progression, and mechanistic insight into how ALK4 regulates the stability of T β RI and T β RII, via MGAT5 and galectin-3. In terms of whether ALK4 might compete with T β RI and T β RII for the machinery that drives their cell surface localization, given the similarity between ALK4 and T β RI we considered that possibility, however our proteomic data and preliminary data support the upregulation of a broad spectrum of glycosylated cell surface proteins/receptors, including integrins in PANC-1 cells and VEGFR2 in endothelial cells (Figure 6g and Figure L4), making this less likely. This is now discussed.

New Reviewer critique: These observations are not particularly relevant to the proposed working model. With that being said, the mechanistic studies necessary to provide definitive evidence that ALK4 suppresses TGF β signaling and thereby prevents the progression of pancreatic and breast cancers are either unconvincing or entirely lacking. Given that previous studies have identified ALK4 as a receptor for TGF β , conducting thorough mechanistic investigations is crucial for the validity of this study. Additionally, a major issue remains: the inactivation of T β RII has been shown to accelerate pancreatic cancer in the KC mouse model of PDAC. How do the authors reconcile their findings with

We thank the reviewer for their thoughtful comments. We would like to clarify that ALK4 is not thought to serve as a receptor for TGF- β 1, 2 or 3 [24, 25]. ALK4 is a type I receptor for Activin A, Nodal and several GDF ligands in the TGF- β family, as described in our manuscript (line190-191).

Regarding the last point, regulation of TGF- β signaling is complex and context-dependent. Inactivation of T β RII has been shown to accelerate pancreatic tumorigenesis, however prior studies have shown that abrogating TGF- β signaling with small molecule inhibitors of ALK5, led to paradoxical increases in TGF- β signaling resulting in cardiac valvulopathy *in vivo* [26]. In addition, our working model posits a distinct mechanism: the loss of ALK4 function does not directly abrogate canonical TGF- β signaling, but rather increase it indirectly. The effects of this modulation on counter-regulatory mechanisms in this tightly controlled pathway remain to be investigated.

Reviewer #5

We thank the reviewer for finding that “The authors made substantial improvement in the revised manuscript, and addressed most issues raised previously.”

Concern 1: With regard to the missing mechanism that accounts for the increased Mgat5 transcription/TbR glycosylation due to loss of ALK4, the authors performed some additional experiments to examine noncanonical TGF-b signaling. However, the results were inconclusive and the newly reported link to ETS1 is insufficiently informative. Given that a role of ActRIIB was also ruled out, the authors should acknowledge as such at least in the discussion to leave this issue open for future exploration.

This has been added to discussion as suggested (lines 451-457).

References:

1. Sasai, K., et al., *UDP-GlcNAc concentration is an important factor in the biosynthesis of beta1,6-branched oligosaccharides: regulation based on the kinetic properties of N-acetylglucosaminyltransferase V*. *Glycobiology*, 2002. **12**(2): p. 119-27.
2. Lau, K.S. and J.W. Dennis, *N-Glycans in cancer progression*. *Glycobiology*, 2008. **18**(10): p. 750-60.
3. Chen, W.H., et al., *Human pancreatic adenocarcinoma: in vitro and in vivo morphology of a new tumor line established from ascites*. *In Vitro*, 1982. **18**(1): p. 24-34.
4. Loukopoulos, P., et al., *Orthotopic transplantation models of pancreatic adenocarcinoma derived from cell lines and primary tumors and displaying varying metastatic activity*. *Pancreas*, 2004. **29**(3): p. 193-203.
5. Moore, P.S., et al., *Genetic profile of 22 pancreatic carcinoma cell lines. Analysis of K-ras, p53, p16 and DPC4/Smad4*. *Virchows Arch*, 2001. **439**(6): p. 798-802.
6. Caldas, C., et al., *Frequent somatic mutations and homozygous deletions of the p16 (MTS1) gene in pancreatic adenocarcinoma*. *Nat Genet*, 1994. **8**(1): p. 27-32.
7. Huang, L., et al., *Deletion and mutation analyses of the P16/MTS-1 tumor suppressor gene in human ductal pancreatic cancer reveals a higher frequency of*

- abnormalities in tumor-derived cell lines than in primary ductal adenocarcinomas. Cancer Res, 1996. 56(5): p. 1137-41.*
8. Schutte, M., et al., *DPC4 gene in various tumor types. Cancer Res, 1996. 56(11): p. 2527-30.*
 9. Freeman, J.W., C.A. Mattingly, and W.E. Strodel, *Increased tumorigenicity in the human pancreatic cell line MIA PaCa-2 is associated with an aberrant regulation of an IGF-1 autocrine loop and lack of expression of the TGF-beta type RII receptor. J Cell Physiol, 1995. 165(1): p. 155-63.*
 10. Lieber, M., et al., *Establishment of a continuous tumor-cell line (panc-1) from a human carcinoma of the exocrine pancreas. Int J Cancer, 1975. 15(5): p. 741-7.*
 11. Berrozpe, G., et al., *Comparative analysis of mutations in the p53 and K-ras genes in pancreatic cancer. Int J Cancer, 1994. 58(2): p. 185-91.*
 12. Simeone, D.M., T. Pham, and C.D. Logsdon, *Disruption of TGFbeta signaling pathways in human pancreatic cancer cells. Ann Surg, 2000. 232(1): p. 73-80.*
 13. Aubert, M., et al., *Decrease of human pancreatic cancer cell tumorigenicity by alpha1,3galactosyltransferase gene transfer. Int J Cancer, 2003. 107(6): p. 910-8.*
 14. Miknyoczki, S.J., et al., *The Trk tyrosine kinase inhibitor CEP-701 (KT-5555) exhibits significant antitumor efficacy in preclinical xenograft models of human pancreatic ductal adenocarcinoma. Clin Cancer Res, 1999. 5(8): p. 2205-12.*
 15. Fukasawa, M. and M. Korc, *Vascular endothelial growth factor-trap suppresses tumorigenicity of multiple pancreatic cancer cell lines. Clin Cancer Res, 2004. 10(10): p. 3327-32.*
 16. Kearney, C.J., et al., *SUGAR-seq enables simultaneous detection of glycans, epitopes, and the transcriptome in single cells. Sci Adv, 2021. 7(8).*
 17. Osuka, R.F., T. Yamasaki, and Y. Kizuka, *Structure and function of N-acetylglucosaminyltransferase V (GnT-V). Biochim Biophys Acta Gen Subj, 2024. 1868(11): p. 130709.*
 18. Partridge, E.A., et al., *Regulation of cytokine receptors by Golgi N-glycan processing and endocytosis. Science, 2004. 306(5693): p. 120-4.*
 19. Granovsky, M., et al., *Suppression of tumor growth and metastasis in Mgat5-deficient mice. Nat Med, 2000. 6(3): p. 306-12.*
 20. Guo, H.B., et al., *Specific posttranslational modification regulates early events in mammary carcinoma formation. Proc Natl Acad Sci U S A, 2010. 107(49): p. 21116-21.*
 21. Carvalho, S., et al., *Preventing E-cadherin aberrant N-glycosylation at Asn-554 improves its critical function in gastric cancer. Oncogene, 2016. 35(13): p. 1619-31.*
 22. Guo, H.B., et al., *Regulation of homotypic cell-cell adhesion by branched N-glycosylation of N-cadherin extracellular EC2 and EC3 domains. J Biol Chem, 2009. 284(50): p. 34986-97.*
 23. Chaney, W., et al., *The Lec4A CHO glycosylation mutant arises from miscompartmentalization of a Golgi glycosyltransferase. J Cell Biol, 1989. 109(5): p. 2089-96.*
 24. Goebel, E.J., et al., *Structures of activin ligand traps using natural sets of type I and type II TGFbeta receptors. iScience, 2022. 25(1): p. 103590.*

25. Moustakas, A. and C.H. Heldin, *The regulation of TGFbeta signal transduction*. Development, 2009. **136**(22): p. 3699-714.
26. Stauber, A.J., et al., *Nonclinical Safety Evaluation of a Transforming Growth Factor β Receptor I Kinase Inhibitor in Fischer 344 Rats and Beagle Dogs*. J Clin Pract, 2014. **4**(3).

REVIEWERS' COMMENTS

Reviewer #1 (Remarks to the Author):

All major issues have been addressed. Two minor presentation/labelling issues are still recommended to be fixed prior to publication

Concern 10 - again strongly recommend that all blots (other than shift in bands) be presented with quantitation.

Thank you, all blots have been labeled with quantitation.

Fig 7h,i,j still need to have the 'KO' label specified as 'ALK4 KO'.

The KO have been labeled as ALK4 KO in this figure as requested.

Reviewer #6 (Remarks to the Author):

The authors have sufficiently addressed the comments of Reviewer 4. Because the evidence for a tumor suppressive role for ALK4 in breast cancer is much weaker than in PDAC, the authors should emphasize the limitations of current evidence in support of a tumor suppressive role in that tumor type, despite positive results in breast cancer cell lines.

Thank you for reviewing our manuscript. We have emphasized this limitation as the reviewer recommended in the revised discussion section.